# Cavity induced collective behavior in the polaritonic ground state

Vasil Rokaj[1,2*], Simeon I. Mistakidis[1,2] and H. R. Sadeghpour[1]

**1** ITAMP, Center for Astrophysics | Harvard & Smithsonian,
Cambridge, Massachusetts 02138, USA
**2** Department of Physics, Harvard University, Cambridge, Massachusetts 02138, USA

⋆ vasil.rokaj@cfa.harvard.edu

## Abstract

Cavity quantum electrodynamics provides an ideal platform to engineer and control light-matter interactions with polariton quasiparticles. In this work, we investigate collective phenomena in a system of many particles in a harmonic trap coupled to a homogeneous cavity vacuum field. The system couples collectively to the cavity field, through its center of mass, and collective polariton states emerge. The cavity field mediates pairwise long-range interactions and enhances the effective mass of the particles. This leads to an enhancement of localization in the matter ground state density, which features a maximum when light and matter are on resonance, and demonstrates a Dicke-like, collective behavior with the particle number. The light-matter interaction also modifies the photonic properties of the polariton system, as the ground state is populated with bunched photons. In addition, it is shown that the diamagnetic $A^2$ term is necessary for the stability of the system, as otherwise the superradiant ground state instability occurs. We demonstrate that coherent transfer of polaritonic population is possible with an external magnetic field and by monitoring the Landau-Zener transition probability.

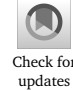

# 1 Introduction

Cavity quantum electrodynamics (cQED) is a rapidly evolving field, combining several different platforms for the manipulation and control of quantum matter with the aid of electromagnetic fields [1–3]. Its range of applicability spans from quantum optics [4], to polaritonic chemistry [5–13], as well as to ultra-cold gases in cavities [14], and light-induced states of matter using either classical [15, 16] or quantum cavity fields [17–20]. In addition, harmonically trapped cold ions in optical cavities have been explored for quantum information processing [21–23] as well as, for light-matter interactions at the single particle level [24–27].

In the last decade, there has been an intense interest on strong and ultrastrong light-matter interactions [28, 29], where light and matter entangle forming hybrid quasiparticle states known as polaritons [28, 30, 31]. Polaritons exhibit remarkable properties which have been probed in condensed matter [2], chemistry [1, 3, 32] and cold-atom [14] settings. They can lead, for instance, to modifications of chemical reactions [5, 7–12, 33], and to the control of excitons while also exciton-polariton condensation has been achieved [34–37]. Moreover, it has been suggested that strong light-matter interactions can influence the electron-phonon coupling and the critical temperature of superconductors [38–42]. Implications for coupling to chiral electromagnetic fields are currently under investigation [17, 43–46], and cavity-induced ferroelectric phases have been proposed [47, 48]. Landau levels in quantum Hall systems exhibit ultrastrong coupling to the cavity field [49–55] and Landau polariton states have been observed [49–51, 56–58]. More recently, theoretical mechanisms on how to modify the integer Hall effect via the cavity field were proposed [59, 60] and the breakdown of the topological protection of the integer Hall effect due to the cavity was demonstrated experimentally [61].

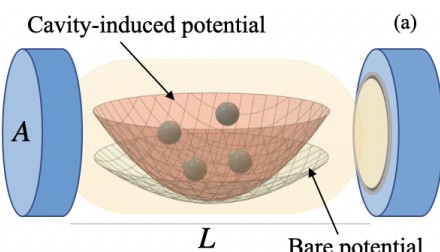
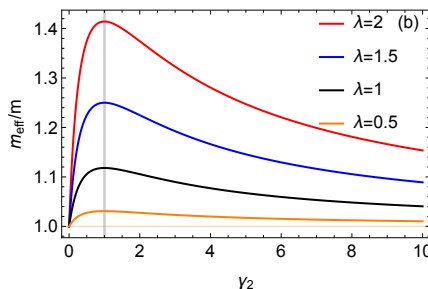

Figure 1: (a) Schematic representation of the interacting system confined in the harmonic potential with frequency $\Omega$ (bare potential in cyan) inside the cavity. The cavity field is considered homogeneous (depicted in light yellow), the area of the cavity mirrors is $A$, the distance among the mirrors is $L$ and the fundamental cavity frequency $\omega = \pi c/L$. The impact of the cavity to the matter field can be effectively understood as a modified (in frequency) harmonic trap and the induction of pairwise long-range interactions. (b) Effective mass ratio $m_{\text{eff}}/m$ as a function of the relative cavity frequency $\gamma_2 = \omega/\Omega$ for different values of the light-matter coupling $\lambda$. Strikingly, the effective mass increases with respect to $\gamma_2$, it maximizes at resonance $\gamma_2 = 1$ and for $\gamma_2 > 1$ it decreases reaching asymptotically its original value, $m_{\text{eff}} \rightarrow m$ for $\gamma_2 \rightarrow \infty$.

All these exciting developments call for further investigations on the properties of many-body systems strongly coupled to the quantized cavity field. In this context, first-principle approaches have been put forward, such as the exact density-functional reformulation of QED (QEDFT) [62–64], hybrid-orbital approaches [65], or generalized coupled cluster theory for analyzing polaritonic phenomena arising in light-matter systems [66, 67]. Complementary, analytical methods and exactly solvable models have played an important role in the development of many-body physics [68], and are highly desirable in the framework of cQED for understanding the origin of the microscopic mechanisms and induced phenomena of strongly correlated light-matter systems.

Some of the key questions in this field, which have been raised in part due to experimental observations, relate to the impact of polariton formation in the ground state of a many-particle system [2, 3, 32, 69]. For example: (i) Can the cavity modify the matter ground state? (ii) Are there collective phenomena in the ground state? (iii) How the resonance between light and matter manifests in the quasiparticle properties? (iv) What are the induced interactions in the matter subsystem due to the light field and the matter-mediated correlations between photons? (v) How can the polaritons be controlled with external probes?

To address the above questions and provide analytical insights into light-matter correlated phenomena, we study a system of many particles where the formation and properties of the polariton states is obtained analytically. As depicted in Fig. 1(a) our system consists of $N$ interacting particles in a harmonic trap embedded in a cavity, whose quantized field is treated in the long-wavelength limit [4, 70, 71]. The particles are considered to be structureless rendering our model exact for electrons, but also applicable to trapped cold ions [72]. Importantly, we show that due to the homogeneity of the cavity field in the long-wavelength limit, the light-matter correlations are solely associated with the center of mass (CM) of the particles.

The ground state properties of the light-matter system are studied and we find that the polariton ground state is populated with photons [73] which obey super-Poissonian statistics, i.e., they bunch [74–76]. Further, we show that the cavity field enhances the localization of the CM wave function which is a consequence of the increasing effective mass of the matter

system. The enhancement of localization becomes maximal when the bare matter and cavity excitations are on resonance as it can also be inferred from the effective mass in Fig. 1(b). This finding highlights the importance of resonance in the ground state of the matter subsystem, and could potentially provide insights into the resonance effect observed in polaritonic chemistry [5–7, 32, 77, 78]. The significance of the CM motion in polaritonic chemistry has also been investigated for charged molecules coupled to a cavity [79]. The effect of the cavity on the matter subsystem can be also understood in terms of an effective Hamiltonian, including solely the matter degrees of freedom, where the particles interact through a cavity induced potential [see Fig. 1(a)]. This description allows to perform many-body treatments avoiding the complication of explicitly including the photon states. Using the effective Hamiltonian and focusing, for simplicity, in the case of no matter interactions, we find that the photon-mediated interactions imprint a weak enhancement of density localization in the matter subsystem [see Fig. 5(a)]. The modification of the density depends on the number of particles $N$ demonstrating a collective behavior. The peak of the density scales with $N$, depending on the light-matter interaction. For weak coupling, it scales proportionally to $N$, while for strong coupling, the behavior is $\sim \sqrt{N}$ [see Fig. 5(b)]. This generalizes Dicke collectivity from being an excited state property of spontaneous emission (superradiance) [80] to a ground state phenomenon as well.

Moreover, we demonstrate that including the diamagnetic $\mathbf{A}^2$ term prevents the system from developing a superradiant instability. This refers to the situation where the ground state becomes unstable and the photon occupation diverges [81,82]. Proceeding one step further we showcase that the collective polariton states can be controlled with the use of a weak external magnetic field. In particular, there is an efficient inter-polariton exchange of energy and control of the polariton gap which is minimized away from the resonant point. This directly impacts the probability of the relevant Landau-Zener transition [83], which increases when the light and matter excitations are out of resonance.

This work proceeds as follows. Sec. 2 introduces the many-particle model coupled to the cavity and demonstrates the separation of the relative coordinates from the cavity field. In Sec. 3 the exact solution for the collective polariton states is outlined. Sec. 4 elaborates on the collective density modifications due to the cavity and the resonance dependence of the effective mass. Sec. 5 discusses the photonic properties of the polariton ground state, while in Sec. 6 we argue on the importance of the diamagnetic interactions regarding the stability of the system. The impact of a weak external magnetic field on the polariton states is appreciated in Sec. 7. In Sec. 8 we draw our conclusions and provide future perspectives.

## 2 Many Particles in a Cavity

We consider a system of $N$ interacting particles, confined in a harmonic potential and coupled to the quantized cavity field. Such a system in the non-relativistic limit is described by the minimal-coupling Hamiltonian [1, 4, 84]

$$\hat{H} = \frac{1}{2m} \sum_{i=1}^{N} \left( i\hbar\nabla_i + g_0\hat{\mathbf{A}} \right)^2 + \sum_{i<l}^{N} W(|\mathbf{r}_i - \mathbf{r}_l|) + \sum_{i=1}^{N} \frac{m\Omega^2}{2}\mathbf{r}_i^2 + \sum_{\boldsymbol{\kappa},\nu} \hbar\omega(\boldsymbol{\kappa}) \left( \hat{a}^\dagger_{\boldsymbol{\kappa},\nu} \hat{a}_{\boldsymbol{\kappa},\nu} + \frac{1}{2} \right), \quad (1)$$

where $g_0$ is the single-particle coupling parameter to the quantized field $\hat{\mathbf{A}}$, which is given in units of the elementary charge, $e$. Further, $m$ is the mass of the particles and $\Omega$ is the frequency of the harmonic trap. Such a model can be realized experimentally with the use of quadrupole ion traps [85], and these systems coupled to optical cavities at the single particle level have been studied experimentally [21–27]. The quantized electromagnetic vector potential $\hat{\mathbf{A}}$ in the

long-wavelength limit (homogeneous approximation) reads [4, 29, 70, 84]

$$\hat{\mathbf{A}} = \sum_{\boldsymbol{\kappa}, \nu} \sqrt{\frac{\hbar}{2\epsilon_0 \mathcal{V} \omega(\boldsymbol{\kappa})}} \, \boldsymbol{\varepsilon}_\nu(\boldsymbol{\kappa}) \left( \hat{a}_{\boldsymbol{\kappa}, \nu} + \hat{a}_{\boldsymbol{\kappa}, \nu}^\dagger \right). \tag{2}$$

Here, $\boldsymbol{\kappa} = (\kappa_x, \kappa_y, \kappa_z)$ are the wave vectors of the photon field, $\omega(\boldsymbol{\kappa}) = c|\boldsymbol{\kappa}|$ are the allowed frequencies in the quantization volume $\mathcal{V}$, $\epsilon_0$ is the vacuum permittivity and $\nu = 1, 2$ denote the two transversal polarization directions [84, 86]. The polarization vectors satisfy $\boldsymbol{\varepsilon}_\nu(\boldsymbol{\kappa}) \cdot \boldsymbol{\kappa} = 0 \;\; \forall \boldsymbol{\kappa}$, and can be taken to be mutually perpendicular $\boldsymbol{\varepsilon}_\nu(\boldsymbol{\kappa}) \cdot \boldsymbol{\varepsilon}_{\nu'}(\boldsymbol{\kappa}) = \delta_{\nu\nu'}$. The operators $\hat{a}_{\boldsymbol{\kappa}, \nu}$ and $\hat{a}_{\boldsymbol{\kappa}, \nu}^\dagger$ are the annihilation and creation operators of the photon field obeying $[\hat{a}_{\boldsymbol{\kappa}, \nu}, \hat{a}_{\boldsymbol{\kappa}', \nu'}^\dagger] = \delta_{\boldsymbol{\kappa}\boldsymbol{\kappa}'} \delta_{\nu\nu'}$. The photon operators can also be defined in terms of the displacement coordinates $q_{\boldsymbol{\kappa}, \nu}$ and their conjugate momenta $\partial / \partial q_{\boldsymbol{\kappa}, \nu}$ as $\hat{a}_{\boldsymbol{\kappa}, \nu} = \frac{1}{\sqrt{2}} \left( q_{\boldsymbol{\kappa}, \nu} + \partial / \partial q_{\boldsymbol{\kappa}, \nu} \right)$ and $\hat{a}_{\boldsymbol{\kappa}, \nu}^\dagger$ defined by conjugation [84, 87].

## 2.1 Kinematics of decoupling of center of mass and relative coordinates

Upon expanding the covariant kinetic energy term it can be seen that the homogeneous photon field couples to the total momentum of the particles, $\hat{\mathbf{A}} \cdot \sum_{i=1}^{N} \nabla_i$, implying that the particles couple collectively to the cavity field through the CM. Consequently, it is beneficial to transform the Hamiltonian into the CM frame and relative coordinates for describing properly the matter-photon interaction and correlations. Here, for mathematical convenience, we utilize a symmetric definition with respect to $\sqrt{N}$ as in Ref. [88]

$$\mathbf{R} = \frac{1}{\sqrt{N}} \sum_{i=1}^{N} \mathbf{r}_i \quad \text{and} \quad \mathbf{R}_j = \frac{\mathbf{r}_1 - \mathbf{r}_j}{\sqrt{N}}, \qquad \text{for } j > 1. \tag{3}$$

As expected, the respective relative and CM conjugate operators commute, demonstrating the independence of position and momentum coordinates.[1] In the new coordinates, the cavity field couples only to the CM momentum, $\hat{\mathbf{A}} \cdot \sum_{i=1}^{N} \nabla_i = \sqrt{N} \hat{\mathbf{A}} \cdot \nabla_{\mathbf{R}}$, while the scalar trapping potential separates into two independent parts, one depending on the CM coordinate and the other depending on the relative coordinates, without any crossing terms between them, $\sum_{i=1}^{N} \mathbf{r}_i^2 = \mathbf{R}^2 + N \sum_{j=2}^{N} \mathbf{R}_j^2 - \left( \sum_{j=2}^{N} \mathbf{R}_j \right)^2$. The two-body interaction $W(|\mathbf{r}_i - \mathbf{r}_l|)$ depends only on the relative distances and thereby does not affect the cavity-induced CM motion. The Hamiltonian therefore has two parts: (i) the CM contribution via $\hat{H}_{\text{cm}}$ which couples to the quantized field $\hat{\mathbf{A}}$ and (ii) the relative contribution $\hat{H}_{\text{rel}}$ being decoupled from both the cavity field $\hat{\mathbf{A}}$ and the CM. As such, $\hat{H} = \hat{H}_{\text{cm}} + \hat{H}_{\text{rel}}$, where

$$\hat{H}_{\text{cm}} = \frac{1}{2m} \left( i\hbar \nabla_{\mathbf{R}} + g_0 \sqrt{N} \hat{\mathbf{A}} \right)^2 + \frac{m\Omega^2}{2} \mathbf{R}^2 + \sum_{\boldsymbol{\kappa}, \nu} \hbar\omega(\boldsymbol{\kappa}) \left( \hat{a}_{\boldsymbol{\kappa}, \nu}^\dagger \hat{a}_{\boldsymbol{\kappa}, \nu} + \frac{1}{2} \right). \tag{4}$$

For the expression of $\hat{H}_{\text{rel}}$ see Appendix A. Therefore, in the presence of a homogeneous cavity field consisting of an arbitrary amount of modes, and harmonically trapped charged particles, the light-matter interaction takes place between the cavity field and the CM. The pairwise interaction drops out because our treatment is for structureless particles and a homogeneous cavity field. This is a generalization of Kohn's theorem [90] which for a homogeneous electron gas in a homogeneous magnetic field, produces CM cyclotron transitions unaffected by two-body interactions. We note however that an inhomogeneous photon field or anharmonic

---

[1]A model of electrons in cavity was recently considered in [89], where a separation of the relative coordinates from the CM and the cavity field was formulated. However, as it was also stated in Ref. [89], the coordinates used therein are linearly dependent and thus the separation of coordinates is not achieved.

trapping potential couples the relative motion to the cavity. In addition, the decoupling of the relative coordinates from the CM and the cavity field holds for an external homogeneous time-dependent classical field $\mathbf{A}_{\text{ext}}(t)$, $\mathbf{E}_{\text{ext}}(t) = -\partial_t \mathbf{A}_{\text{ext}}(t)$.

# 3 Collective Polariton States

We consider now the case of a single-mode cavity, i.e., $\kappa_x = \kappa_y = 0$ while $\kappa_z \neq 0$. Then, the polarization vectors are $\boldsymbol{\varepsilon}_1 = \mathbf{e}_x$ and $\boldsymbol{\varepsilon}_2 = \mathbf{e}_y$ and the field simplifies to

$$\hat{\mathbf{A}} = \sum_{v=x,y} \sqrt{\frac{\hbar}{2\epsilon_0 \mathcal{V} \omega}} \mathbf{e}_v \left( \hat{a}_v + \hat{a}_v^\dagger \right). \tag{5}$$

Since the polarization vectors of the photon field lie in the $(x, y)$ plane, the $z$ direction of the system becomes trivial and can be neglected. The light-matter Hamiltonian then becomes a system of interacting harmonic oscillators

$$\hat{H}_{\text{cm}} = -\frac{\hbar^2}{2m} \nabla_\mathbf{R}^2 + \frac{\mathrm{i} g_0 \hbar}{m} \sqrt{N} \hat{\mathbf{A}} \cdot \nabla_\mathbf{R} + \frac{m\Omega^2}{2} \mathbf{R}^2 + \underbrace{\frac{N g_0^2}{2m} \hat{\mathbf{A}}^2 + \sum_{v=x,y} \hbar\omega \left( \hat{a}_v^\dagger \hat{a}_v + \frac{1}{2} \right)}_{\hat{H}_p}. \tag{6}$$

The Hamiltonian $\hat{H}_p$ includes also the photonic annihilation and creation operators, and it can be brought to the diagonal form of a harmonic oscillator through the scaling transformation $u_v = q_v \sqrt{\widetilde{\omega}/\omega}$ [recall that $\hat{a}_v = (q_v + \partial/\partial q_v)/\sqrt{2}$] and becomes $\hat{H}_p = \sum_v \frac{\hbar\widetilde{\omega}}{2} \left( -\partial^2/\partial u_v^2 + u_v^2 \right)$. The frequency $\widetilde{\omega}$ is the dressed cavity frequency

$$\widetilde{\omega} = \sqrt{\omega^2 + \omega_d^2}, \quad \text{with} \quad \omega_d = \sqrt{\frac{g_0^2 N}{m\epsilon_0 \mathcal{V}}} = \sqrt{\frac{g_0^2 2 n_{2\mathrm{D}} \omega}{m\epsilon_0 \pi c}}. \tag{7}$$

We note that $\omega_d$ is the *diamagnetic frequency* which originates from the $\hat{\mathbf{A}}^2$ term in the Hamiltonian [57, 70, 71, 91–93]. Further, in Eq. (7) we used the expression for the fundamental cavity frequency $\omega = \pi c/L$ and we introduced the 2D particle density $n_{2\mathrm{D}} = N/A$, where $A$ refers to the area of the cavity mirrors and $L$ to the cavity length as shown in Fig. 1(a).

## 3.1 Derivation of the polariton states

After the scaling transformation the full CM Hamiltonian reads

$$\hat{H}_{\text{cm}} = \sum_{v=x,y} \left[ -\frac{\hbar^2}{2m} \frac{\partial^2}{\partial R_v^2} + \frac{m\Omega^2 R_v^2}{2} + \mathrm{i} g \sqrt{2} u_v \frac{\partial}{\partial R_v} + \frac{\hbar\widetilde{\omega}}{2} \left( -\frac{\partial^2}{\partial u_v^2} + u_v^2 \right) \right] \equiv \sum_{v=x,y} \hat{H}_v. \tag{8}$$

The Hamiltonian consists of two copies, namely $\hat{H}_{\text{cm}} = \sum_v \hat{H}_v$. Without loss of generality we focus on a single copy, i.e. $\hat{H}_v$. To avoid any confusion we note that $\mathbf{R} = (R_x, R_y) = (X, Y)$ and $\nabla = (\partial_x, \partial_y)$. Further, the collective light-matter coupling $g = \omega_d \sqrt{\hbar^3/2m\widetilde{\omega}}$ depends on the particle number $N$ through the diamagnetic frequency $\omega_d$ and whose scaling $g \sim \sqrt{N}$ is the same as in the few-level models of quantum optics [28, 80, 94, 95]. Here, this occurs naturally due to the collective CM excitation that couples to the cavity. The coupling between the two oscillators can be brought into a coordinate-coordinate form (facilitating its analytical treatment) through the Fourier transform, $\phi(R_v) = \int_{-\infty}^{\infty} \frac{dK_v}{2\pi} \tilde{\phi}(K_v) e^{\mathrm{i} K_v R_v}$, see details in Ref. [96],

$$\hat{H}_v = -\frac{m\Omega^2}{2} \frac{\partial^2}{\partial K_v} + \frac{\hbar^2}{2m} K_v^2 - g\sqrt{2} K_v u_v + \frac{\hbar\widetilde{\omega}}{2} \left[ -\frac{\partial^2}{\partial u_v^2} + u_v^2 \right]. \tag{9}$$

We introduce the scaled coordinates $V_{\nu+} = K_\nu \sqrt{\hbar^2/m\Omega^2}$ and $V_{\nu-} = -u_\nu \sqrt{\hbar/\tilde{\omega}}$ in order to bring the Hamiltonian into the form of two interacting harmonic oscillators with unit mass $\hat{H}_\nu = -\frac{\hbar^2}{2} \sum_{l=\pm} \frac{\partial^2}{\partial V_{\nu l}^2} + \frac{1}{2} \sum_{l,j=\pm} W_{lj} V_{\nu l} V_{\nu j}$ where the elements of W are $W_{++} = \Omega^2$, $W_{--} = \tilde{\omega}^2$ and $W_{+-} = W_{-+} = \omega_d \Omega$ and thus the matrix $W$ is real and symmetric. As a consequence it can be diagonalized by the orthogonal matrix $O$ [71],

$$O = \begin{pmatrix} \frac{1}{\sqrt{1+\Lambda^2}} & \frac{\Lambda}{\sqrt{1+\Lambda^2}} \\ -\frac{\Lambda}{\sqrt{1+\Lambda^2}} & \frac{1}{\sqrt{1+\Lambda^2}} \end{pmatrix}, \quad \text{with} \quad \Lambda = \alpha - \sqrt{1+\alpha^2}, \tag{10}$$

and $\alpha = (\Omega^2 - \tilde{\omega}^2)/2\omega_d\Omega$. The parameter $\Lambda$ quantifies how much the matrix $O$ deviates from being diagonal. The Hamiltonian after the orthogonal transformation takes the standard canonical form,

$$\hat{H}_\nu = -\frac{\hbar^2}{2} \sum_{l=\pm} \frac{\partial^2}{\partial S_{\nu l}^2} + \frac{1}{2} \sum_{l=\pm} \Omega_l^2 S_{\nu l}^2, \tag{11}$$

and we obtain the polariton modes, $\Omega_\pm^2 = \frac{1}{2}\left(\tilde{\omega}^2 + \Omega^2 \pm \sqrt{4\omega_d^2\Omega^2 + (\tilde{\omega}^2 - \Omega^2)^2}\right)$. The new coordinates $S_{\nu l}$ and conjugate momenta $\partial_{S_{\nu l}}$ are related to the old ones $\{V_{\nu l}, \partial_{V_{\nu l}}\}$ through the orthogonal matrix $O$ [71], $S_{\nu l} = \sum_j O_{jl} V_{\nu j}$ and $\partial/\partial S_{\nu l} = \sum_j O_{jl} \partial/\partial V_{\nu j}$. $\Lambda$ is the mixing parameter between the matter and the photonic degrees of freedom. Due to the fact that the matrix $O$ is orthogonal the canonical commutation relations are satisfied which implies that we have two independent harmonic oscillators [71]. Thus, the polariton eigenfunctions of the system are Hermite functions $\phi$ of coordinates $S_{\nu+}$ and $S_{\nu-}$

$$\Psi_{n_+,n_-}(S_{\nu+}, S_{\nu-}) = \phi_{n_+}(S_{\nu+}) \otimes \phi_{n_-}(S_{\nu-}), \quad n_\pm \in \mathbb{N}, \tag{12}$$

with eigenspectrum $E_{n_+,n_-} = \hbar\Omega_+(n_+ + 1/2) + \hbar\Omega_-(n_- + 1/2)$. Moreover, it is useful to express the diagonalized Hamiltonian $\hat{H}_{\text{cm}}$ in terms of polaritonic annihilation, $\hat{d}_{\nu l} = S_{\nu l}\sqrt{\frac{\Omega_l}{2\hbar}} + \sqrt{\frac{\hbar}{2\Omega_l}}\partial_{S_{\nu l}}$, and creation, $\hat{d}_{\nu l}^\dagger = S_{\nu l}\sqrt{\frac{\Omega_l}{2\hbar}} - \sqrt{\frac{\hbar}{2\Omega_l}}\partial_{S_{\nu l}}$, operators [87] as $\hat{H}_\nu = \sum_{l=\pm} \hbar\Omega_l\left(\hat{d}_{\nu l}^\dagger \hat{d}_{\nu l} + 1/2\right)$. We note that the polariton eigenstates $\Psi_{n_+,n_-}(S_{\nu+}, S_{\nu-})$ can also be written as Fock states $\Psi_{n_+,n_-}(S_{\nu+}, S_{\nu-}) \equiv |n_+\rangle_\nu |n_-\rangle_\nu$, which can be constructed by applying the polariton creation operators $\hat{d}_{\nu+}^\dagger$ and $\hat{d}_{\nu-}^\dagger$ on the polaritonic vacuum states $|0_+\rangle_\nu |0_-\rangle_\nu$ [86,87].

## 3.2 Tunability of the polariton branches and limiting cases

*Decoupling limit.* The light-matter interaction in our system is controlled by the diamagnetic frequency $\omega_d$. As $\omega_d \to 0$ the polariton modes become $\Omega_\pm^2 = \frac{1}{2}\left(\omega^2 + \Omega^2 \pm \sqrt{(\omega^2 - \Omega^2)^2}\right)$. When $\Omega > \omega$, the upper polariton branch tends to $\Omega_+ \to \Omega$, while the lower polariton approaches $\Omega_- \to \omega$. For $\omega > \Omega$, the situation is inverted, namely $\Omega_+ \to \omega$ and $\Omega_- \to \Omega$. Thus, the correct decoupling limit is consistently recovered.

*No-trap limit.* In the case of a vanishing external trap, i.e. $\Omega \to 0$, the upper (lower) polariton branch approaches the dressed cavity frequency (zero), namely $\Omega_+ \to \tilde{\omega}$ ($\Omega_- \to 0$). This is consistent with the free particle solution in the cavity, where there is only one discrete quantized mode in the system, $\tilde{\omega}$, as it was demonstrated in Ref. [73] meaning that one part of the spectrum becomes continuous [73]. For more details see Appendix B.

*Tuning parameters.* The polaritonic branches [see Eq. (11) and below] are tunable through: (i) the particle number (or particle density) appearing in the diamagnetic frequency $\omega_d$ and (ii) the cavity frequency $\omega$ which can be varied by changing the distance $L$ among the cavity mirrors. The normalized polariton modes in terms of the trap frequency $\Omega$ read

$$\frac{\Omega_\pm}{\Omega} = \sqrt{\frac{\gamma_1^2 + \gamma_2^2 + 1 \pm \sqrt{4\gamma_1^2 + \left(\gamma_1^2 + \gamma_2^2 - 1\right)^2}}{2}}, \tag{13}$$

where $\gamma_1 = \omega_d/\Omega$ and $\gamma_2 = \omega/\Omega$ are dimensionless ratios. Notice also that the mixing parameter $\Lambda = \alpha - \sqrt{1+\alpha^2}$, defined in Eq. (10), can be equivalently written solely in terms of $\gamma_1$ and $\gamma_2$ as $\alpha = (\gamma_1^{-1} - \gamma_1 - \gamma_2^2\gamma_1^{-1})/2$. Notably, the dimensionless ratios are not independent because the diamagnetic frequency $\omega_d$ is also a function of $\omega$ [see Eq. (7)]. As a consequence, the normalized diamagnetic frequency $\omega_d/\Omega$ can be written as $\gamma_1 = \omega_d/\Omega = \lambda\sqrt{\gamma_2}$. Thus, the two independent dimensionless parameters in our setting are

$$\lambda = \sqrt{\frac{g_0^2 N}{A m \epsilon_0 \pi c \Omega}} \quad \text{and} \quad \gamma_2 = \frac{\omega}{\Omega}. \tag{14}$$

It is evident that the dimensionless light-matter coupling $\lambda$ can be tuned by varying the particle number $N$ and the area of the cavity mirrors $A$. Thus, $\lambda$ can be flexibly adjusted taking a wide range of values which correspond to different light-matter coupling regimes.

## 3.3 Light-matter coupling regimes and polariton behavior

*Light-matter coupling regimes.* The weak coupling regime is dictated by the Purcell effect [97], where there is no hybridization between light and matter. While in the strong coupling regime, hybridization enforces the emergence of the Rabi splitting. Experimentally, the two situations are understood by comparing the losses of the system to the light-matter coupling strength [28]. In contrast, the ultrastrong coupling regime is defined by the dimensionless ratio between the light-matter coupling strength and the bare excitations of the system [28]. Thus, this regime determines whether particular approximations, like the rotating-wave approximation, are applicable [28]. In this regime the Rabi splitting is comparable to the bare system excitations and the counter-rotating and the diamagnetic $\mathbf{A}^2$ terms need to be included [28] as it has been demonstrated also experimentally [51]. Furthermore, the so-called deep strong coupling regime has also been achieved where the ratio between the light-matter coupling and the bare excitations approaches unity [49] or even goes beyond it [52]. We will now study the influence of $\lambda$ and $\gamma_2$ on the polariton modes.

*Polariton behavior.* Let us start by investigating the impact of the light-matter coupling, $\lambda$ on the polariton branches. For simplicity, the cavity frequency $\omega$ (or equivalently $\gamma_2$) is held fixed. The resultant upper and lower polaritons as a function of $\lambda$ are presented in Fig. 2 for two different values of the relative cavity frequency $\gamma_2$. The case of $\gamma_2 = 1/2$ corresponding to the situation where the cavity frequency $\omega$ is off-resonant with the trap frequency $\Omega$ is depicted in Fig. 2(a). The upper polariton branch, $\Omega_+$, increases as a function of $\lambda$ while the lower one decreases approaching zero asymptotically. As expected, in the decoupling limit, $\lambda \to 0$, it holds that $\Omega_+ \to \Omega$ and $\Omega_- \to \omega$. Moreover, we observe that the two polariton branches are always separated and do not coincide even for $\lambda \to 0$ since the cavity and the trap frequencies are off-resonance ($\omega \neq \Omega$). Considering a resonantly coupled cavity with the trap, $\gamma_2 = 1$, [Fig. 2(b)] it is found that $\Omega_+$ increases and $\Omega_-$ decreases as a function of $\lambda$ but with a faster rate in comparison to the off-resonant situation. Due to the resonance condition the polariton gap closes for the light-matter interaction approaching zero, $\lambda \to 0$. It is important to mention that despite the fact that the lower polariton has a smaller value than the upper polariton, the

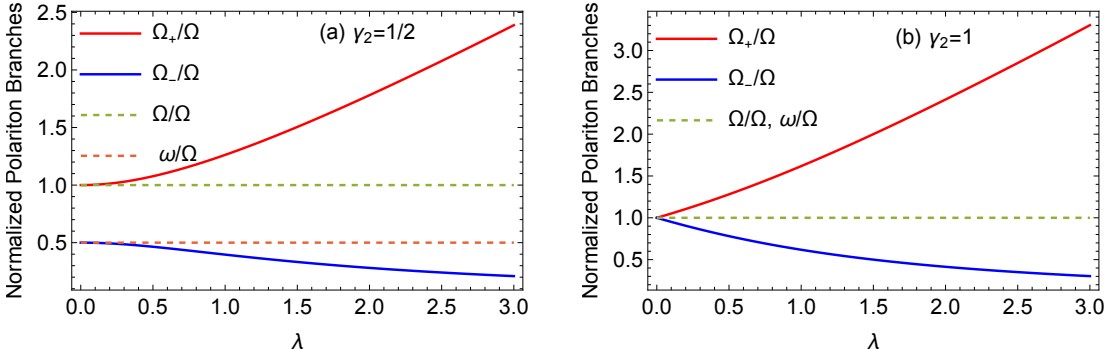

Figure 2: Normalized polariton branches $\Omega_+/\Omega$ and $\Omega_-/\Omega$ (solid lines) as a function of the light-matter coupling $\lambda$. In (a) the cavity and the harmonic trap are off-resonance with $\gamma_2 = \omega/\Omega = 1/2$, while in (b) they are in resonance i.e. $\gamma_2 = \omega/\Omega = 1$. The dashed lines indicate the bare excitation frequencies of the trap and the cavity. In both cases the polariton gap increases for larger $\lambda$.

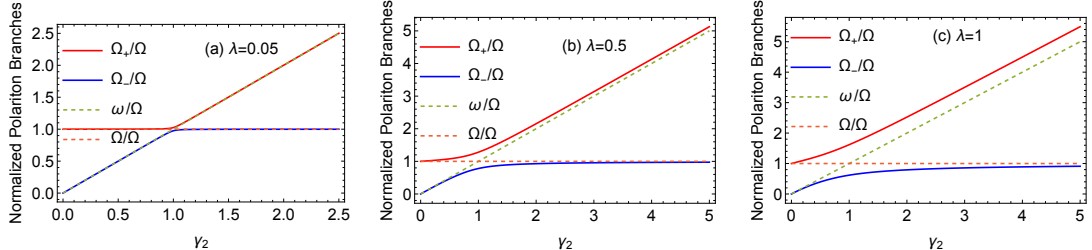

Figure 3: Normalized polariton branches $\Omega_+/\Omega$ and $\Omega_-/\Omega$ (solid lines) as a function of $\gamma_2 = \omega/\Omega$ and fixed values of the light-matter coupling $\lambda$ (see legends). Dashed lines mark the bare excitation frequencies of the matter and the cavity fields. In (a) $\lambda = 0.05$ where an avoided crossing (Rabi splitting) appears at the resonance point $\gamma_2 = 1$. In (b) $\lambda = 0.5$ with the Rabi splitting becoming comparable to the one of bare excitations implying ultrastrong coupling, while in (c) $\lambda = 1$ we enter the deep strong coupling regime in which the upper polariton is parallel to the photon excitation without reaching it.

former is actually more important for the low energy physics of the system. This is especially the case in the ultrastrong and the deep strong coupling regimes since then the energy gap between the two polaritons will be filled with multiple of the excited states ($n_- > 0$) of the lower polariton.

Next, we study the behavior of the polaritons as a function of the relative cavity frequency $\gamma_2 = \omega/\Omega$ for different values of $\lambda$. Fig. 3(a) illustrates the normalized polariton branches $\Omega_\pm/\Omega$ for $\lambda = 0.05$ where it becomes evident that the light and the matter excitations hybridize and an avoided crossing takes place at the resonance point $\gamma_2 = 1$. This signifies the strong coupling between light and matter. Before and after $\gamma_2 = 1$ the polaritonic excitations lie on top of the bare system excitations, i.e. $\omega$ and $\Omega$ respectively. Increasing the light-matter coupling by an order of magnitude to $\lambda = 0.5$ as shown in Fig. 3(b), leads to a considerably larger Rabi splitting among the branches. The associated polariton gap is comparable to the bare excitations of the system. This is a manifestation of the ultrastrong coupling regime [28], where the polaritons deviate for a larger range of $\gamma_2$ from the bare excitations as compared to smaller values of $\gamma_2$. Turning to the case at which $\lambda = 1$ [Fig. 3(c)], i.e. bringing the system to the deep strong coupling regime as defined in Ref [28], it becomes apparent that the Rabi

splitting is arguably more pronounced. Also, the upper polariton deviates almost entirely from the bare excitations of the system and it actually does not reach the bare photon excitation for $\gamma_2 > 1$ but goes parallel to it. This phenomenon where the polaritons do not reach the bare excitations when departing from the resonance point has also been reported experimentally in Landau polariton systems [49]. Overall, Figs. 2 and 3 convey that the Rabi splitting can be controlled by the number of particles $N$, which enters the definition of the interaction parameter $\lambda$ [Eq. (14)], as a consequence of the collective coupling of the particles to the cavity mode through the CM. Such a collective (or cooperative) behavior has been observed experimentally for magnonic solid-state systems coupled to a photon mode and it was dubbed "Dicke cooperativity" [98].

# 4 Ground State Collectivity and Resonance Effect

## 4.1 Effective mass increase and resonance effect

In what follows we focus on the impact of the light field on the matter subsystem, by examining cavity-modified localization properties of matter and its effective mass. To this end, we first inspect the impact of the light-matter coupling on the ground state density profile of the CM. Utilizing, the polariton coordinates $S_{\nu+} = (V_{\nu+} - \Lambda V_{\nu-})/\sqrt{1+\Lambda^2}$ and $S_{\nu-} = (V_{\nu-} + \Lambda V_{\nu+})/\sqrt{1+\Lambda^2}$, the form of $V_{\nu+} = K_\nu \sqrt{\hbar^2/m\Omega^2}$ as well as $V_{\nu-} = -u_\nu \sqrt{\hbar/\widetilde{\omega}}$, the ground state wave function reads

$$\Psi_{\text{gs}} = \prod_{\nu=x,y} \phi_0(S_{\nu+}) \otimes \phi(S_{\nu-}) = \prod_{\nu=x,y} e^{\frac{-\Omega_+\left(K_\nu\sqrt{\hbar^2/m\Omega^2} + \Lambda u_\nu \sqrt{\hbar/\widetilde{\omega}}\right)^2}{2\hbar(1+\Lambda^2)}} e^{\frac{-\Omega_-\left(\Lambda K_\nu\sqrt{\hbar^2/m\Omega^2} - u_\nu \sqrt{\hbar/\widetilde{\omega}}\right)^2}{2\hbar(1+\Lambda^2)}}, \quad (15)$$

where, for simplicity, we have omitted the normalization constant. For the density profile we express the wave function in real space through a Fourier transform, and integrate out the photonic coordinates $u_\nu$. This leads to the CM density profile

$$n_{\text{cm}}(\mathbf{R}) = \frac{|\Psi_{gs}(\mathbf{R})|^2}{|\Psi_{gs}(0)|^2} = \exp\left(-\frac{m_{\text{eff}}\Omega\mathbf{R}^2}{\hbar}\right), \quad \text{with} \quad m_{\text{eff}} = m\frac{1+\Lambda^2}{\Omega_+/\Omega + \Lambda^2\Omega_-/\Omega}. \quad (16)$$

The density profile of the CM, $n_{\text{cm}}(\mathbf{R})$, has a Gaussian form which is modified by the effective mass parameter $m_{\text{eff}}$ that depends on the polariton modes $\Omega_+$, $\Omega_-$ and the mixing parameter $\Lambda$. To appreciate this phenomenon we look at the full-width-at-half-maximum (FWHM), FWHM $= 2\sqrt{2\ln 2}\sigma$, which is proportional to the standard deviation $\sigma$ and thus quantifies the spatial localization of the CM density profile. In particular, for $n_{\text{cm}}(\mathbf{R})$ under the influence of the cavity field, $\sigma = \sqrt{\hbar/2m_{\text{eff}}\Omega}$, while in the case of no cavity it is $\sigma_0 = \sqrt{\hbar/2m\Omega}$. Therefore, FWHM/FWHM$_0 = \sigma/\sigma_0 = \sqrt{m/m_{\text{eff}}}$ which means that the cavity-effect on the CM density profile is reflected in the relative effective mass $m_{\text{eff}}/m$.

In Figure 4 we examine the response of $m_{\text{eff}}/m$ for varying $\lambda$ and $\gamma_2 = \omega/\Omega$. It is observed that independently of $\gamma_2$, $m_{\text{eff}}/m$ increases for larger $\lambda$. This behavior is a direct consequence of the light induced dressing to the matter field. Especially, in the region $0 < \lambda < 2$ the effective mass features a quadratic increase with respect to $\lambda$, and beyond this region $m_{\text{eff}}$ grows in a linear fashion. The rate of increase of $m_{\text{eff}}$ is determined by $\gamma_2$, see also Fig. 1(b). Interestingly, at resonance $\gamma_2 = 1$ the effective mass experiences a relatively faster increase as compared to other values of $\gamma_2$ and also its magnitude is maximized. This implies that when the cavity is at resonance with the trap frequency the dressing of matter by the cavity photons is maximized. In the decoupling limit ($\lambda \to 0$) it holds that $m_{\text{eff}} = m$, as expected due to vanishing dressing. Nevertheless, the impact of finite $\lambda$ on $m_{\text{eff}}$ is arguably noticeable

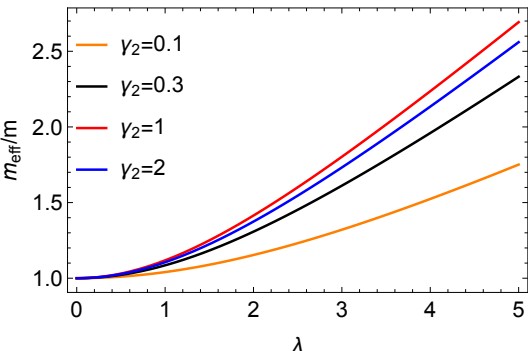

Figure 4: Effective mass ratio $m_{\text{eff}}/m$ as a function of the light-matter coupling $\lambda \sim \sqrt{N}$ for fixed $\gamma_2 = \omega/\Omega$ (see legend). The effective mass becomes larger with increasing $\lambda$, while it features its maximal enhancement at resonance, i.e. for $\gamma_2 = 1$.

especially for $\lambda > 1$, a result that facilitates its experimental detection in contrast, for instance, to a phononic dressing cloud [99, 100]. To understand better the resonance effect we provide in Fig. 1(b) $m_{\text{eff}}/m$ in terms of $\gamma_2 = \omega/\Omega$ for fixed values of $\lambda$. We observe that for all $\lambda$ the effective mass grows rapidly in the region $0 \leq \gamma_2 \leq 1$, it reaches a maximum at resonance $\gamma_2 = 1$ and afterwards decreases approaching asymptotically its bare value, $m_{\text{eff}} \to m$. This means that when $\omega \gg \Omega$ the matter subsystem experiences a gradually lesser influence by the cavity field. This resonance phenomenon imprinted as a maximization of the effective mass could potentially provide insights with respect to the resonance effect that is observed in polaritonic chemistry. In this context, alterations of the chemical reactions and properties, depend crucially on the resonance between the vibronic excitations and the cavity mode [5–7, 32, 77, 78].

## 4.2 Cavity Induced Effective Matter Hamiltonian

The fact that the CM density gets modified by the cavity can be also understood in terms of an effective potential that the CM experiences due to the presence of the cavity. This is a common approach in quasiparticle physics [101–103]. Indeed, besides the external trap $V(\mathbf{R}) = m\Omega^2\mathbf{R}^2/2$, due to the cavity the CM feels the effective potential $V_{\text{eff}}(\mathbf{R}) = m_{\text{eff}}\Omega^2\mathbf{R}^2/2$. This scalar potential has precisely the same effect on the CM density profile as the effect described by Eq. (16). Thus, the cavity-mediated potential $V_{\text{cav}}(\mathbf{R}) = V_{\text{eff}}(\mathbf{R}) - V(\mathbf{R}) = \delta m\Omega^2\mathbf{R}^2/2$ where $\delta m = m_{\text{eff}} - m$, introduces a modified harmonic trap and most importantly an additional bilinear, long-range, interparticle interaction. Similar cavity mediated two-body interactions have also been studied in [104]. This can be directly seen by expanding $V_{\text{cav}}$ in terms of the original single-particle coordinates $\{\mathbf{r}_i\}$

$$V_{\text{cav}}(\mathbf{r}_i, \mathbf{r}_j) = \frac{\delta m\Omega^2}{2N}\left[\sum_{i=1}^{N}\mathbf{r}_i^2 + 2\sum_{i<j}^{N}\mathbf{r}_i \cdot \mathbf{r}_j\right]. \tag{17}$$

Thus, it is possible to construct an effective purely matter Hamiltonian for the description of harmonically trapped many-body systems strongly coupled to a cavity which reads

$$\hat{H}_{\text{eff}} = \sum_{i=1}^{N}\left[-\frac{\hbar^2}{2m}\nabla_i^2 + \left(m + \frac{\delta m}{N}\right)\frac{\Omega^2\mathbf{r}_i^2}{2}\right] + \sum_{i<l}^{N}\left[W(|\mathbf{r}_i - \mathbf{r}_l|) + \frac{\delta m\Omega^2}{N}\mathbf{r}_i \cdot \mathbf{r}_l\right]. \tag{18}$$

The Hamiltonian $\hat{H}_{\text{eff}}$ allows to describe the ground state properties of the matter subsystem under the influence of a cavity, because it captures the fundamental localization of the CM wave function, without the need to account for the photonic states. Similar "photon-free" approaches to light-matter interactions have also been put forward recently in QEDFT [105], but also in effective theories for light-matter interactions [106]. It is important to mention that $V_{\text{cav}}(\mathbf{r}_i, \mathbf{r}_j)$ includes the resonant effect originating from the dependence of $\delta m$.

## 4.3 Collective Effect in the Ground State Density

Having obtained the cavity-induced interactions our aim now is to understand the impact of $V_{\text{cav}}(\mathbf{r}_i, \mathbf{r}_j)$ in the ground state of the trapped matter subsystem, with a particular emphasis on the transition from small to large particle numbers. This will allow us to address whether collectivity emerges due to the cavity-induced interactions. Since we are interested in the effect of $V_{\text{cav}}(\mathbf{r}_i, \mathbf{r}_j)$, we consider for simplicity harmonically trapped non-interacting bosons, $W(\mathbf{r}_i - \mathbf{r}_j) = 0$, coupled to the cavity. In this limit, the two spatial dimensions ($x$ and $y$) in $\hat{H}_{\text{eff}}$ of Eq.(18) are exactly identical, and thus treating only the $x$ direction is sufficient to understand the underlying physical processes. This fact is also explained for the CM Hamiltonian $\hat{H}_{cm}$ given in Eq.(8). The ground state of the matter system obeys the effective Hamiltonian of Eq. (18), and we use the mean-field approximation with the split-step Crank-Nicolson method [107]. The interplay of particle correlations will be discussed in a future work [108].

The deviations between the ground state density with and without cavity-induced resonant interactions ($\gamma_2 = 1$) are shown in Fig. 5(a) for various particle numbers. It is evident that the cavity enhances the density around the trap center while suppresses it at the edges which clearly indicates an enhanced localization tendency. Therefore, the cavity-mediated interactions modify the density of the matter subsystem in a similar way to the CM wave function. The enhancement in localization is weak, if compared to the bare density. However, this phenomenon can become significant in a multimode cavity [109] which would facilitate its experimental realization. Importantly, these modifications get amplified for larger number of particles as it can be deduced by inspecting the maximal density deviations occurring at $x = 0$ with respect to $N$, see Fig. 5(b). This observation underlines that the enhanced localization of the matter system is a collective, Dicke-type phenomenon in the ground state. This is an important finding as it generalizes Dicke collectivity from being solely an excited state phenomenon [80] to include also the ground state. To the best of our knowledge such a collective ground state behavior has not previously been demonstrated.

We present in Fig. 5(b) the respective density differences at the trap center ($x = 0$) for three distinct single-particle couplings. Notice that for all three values of $\lambda/\sqrt{N}$ (fixed and independent of $N$), the underlying density deviations show qualitatively the same localization behavior due to the cavity (not shown). However, a careful investigation of the peak of the density difference at $x = 0$ and its corresponding scaling behavior reveals that from weak to strong light-matter coupling, there is a power law behavior $N^\beta$ which goes from linear, $\beta = 1$, to a square root behavior, $\beta = 1/2$. This demonstrates the impact of the single-particle coupling on the collective behavior and can be traced back to the scaling of the effective mass, see also Fig. 4. The most pronounced deviations are observed for the largest coupling where the scaling is $\sim \sqrt{N}$. It is an interesting prospect to understand whether these scaling behaviors are related to the different collective emission behaviors, superradiance and subradiance, observed in the Dicke model.

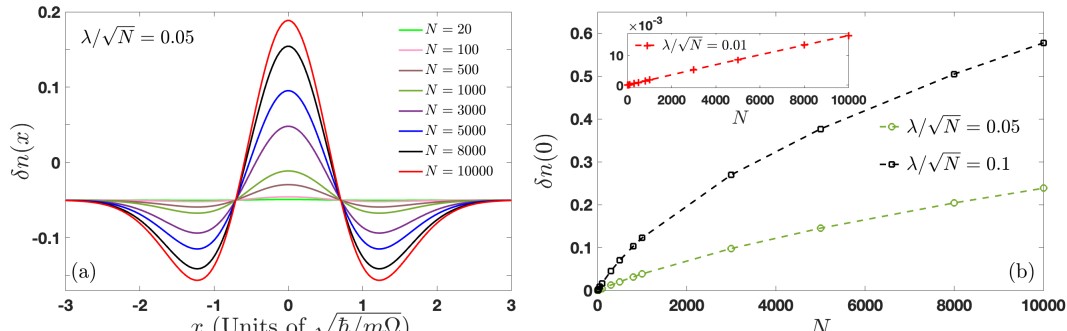

Figure 5: (a) Ground state density difference of the matter subsystem between the coupled and the uncoupled cases for $\lambda/\sqrt{N} = 0.05$ [$\lambda/\sqrt{N}$ is independent of $N$, see Eq. (14)] with respect to the particle number $N$. The cavity mediated interactions increase the density of the coupled system around the trap center signifying enhancement of spatial localization. (b) Density difference at $x = 0$ for three different values of the light-matter coupling as a function of $N$. The density difference demonstrates different scalings revealing the impact of the light-matter coupling on the observed collective behavior.

## 5 Photon Occupations & Photon Correlations

In this section we analyze the photonic properties of the polaritonic ground state, and in particular, the ground state photon occupation and the respective photon correlations. The above will allow us to understand more thoroughly the nature of the polaritonic ground state.

### 5.1 Photon occupations & two-photon processes

First we calculate the photon occupation in the light-matter ground state. The photon operators $\{\hat{a}_\nu, \hat{a}_\nu^\dagger\}$ can be written as combinations of polariton operators $\{\hat{d}_{\nu l}, \hat{d}_{\nu l}^\dagger\}$ (see Appendix C for details). The ground state of the electron-photon system is $|\Psi_{gs}\rangle = \prod_\nu |0_+\rangle_\nu |0_-\rangle_\nu$ which is annihilated by both polariton operators $\hat{d}_{\nu+}$ and $\hat{d}_{\nu-}$, i.e., $\hat{d}_{\nu\pm}|\Psi_{gs}\rangle = 0$. Thus, the photon occupation in the ground state turns out to be

$$\langle \hat{a}_\nu^\dagger \hat{a}_\nu \rangle_{gs} = \frac{1}{4 + 4\Lambda^2}\left[ \frac{\Omega_-/\Omega}{\gamma_2} + \frac{\gamma_2}{\Omega_-/\Omega} - 2 + \Lambda^2\left(\frac{\Omega_+/\Omega}{\gamma_2} + \frac{\gamma_2}{\Omega_+/\Omega} - 2\right)\right], \qquad (19)$$

where the relative polariton modes $\Omega_-/\Omega$, $\Omega_+/\Omega$ and the relative cavity frequency $\gamma_2$ have been introduced. The above result is important as it demonstrates that due to the light-matter interaction there are photons occupying the polaritonic ground state. The amount of photons in the ground state depends crucially on $\Omega_\pm/\omega$. The behavior of the photon occupation in the ground state is shown in Fig. 6(a) as a function of $\lambda$ for different $\gamma_2 = \omega/\Omega$. We observe that in the interval $0 \leq \lambda \leq 2$, the photon occupation increases approximately quadratically in terms of $\lambda$ independently of $\gamma_2$. This fact can be understood from the fit that we perform on the curve characterizing the photon population with $\gamma_2 = 0.1$ in Fig. 6(a). Since the light-matter coupling is proportional to $\sqrt{N}$, the photon occupation is proportional to $\langle \hat{a}_\nu^\dagger \hat{a}_\nu \rangle_{gs} \sim N$, in the region $0 \leq \lambda \leq 2$. Despite this, the photon occupation does not overcome unity, because the proportionality constant in the definition of $\lambda$ [see Eq. (14)] is a small number and the photon occupation per particle is miniscule. This means that there is no macroscopic photon occupation (or superradiant ground state phase), as predicted for the Dicke model [81].

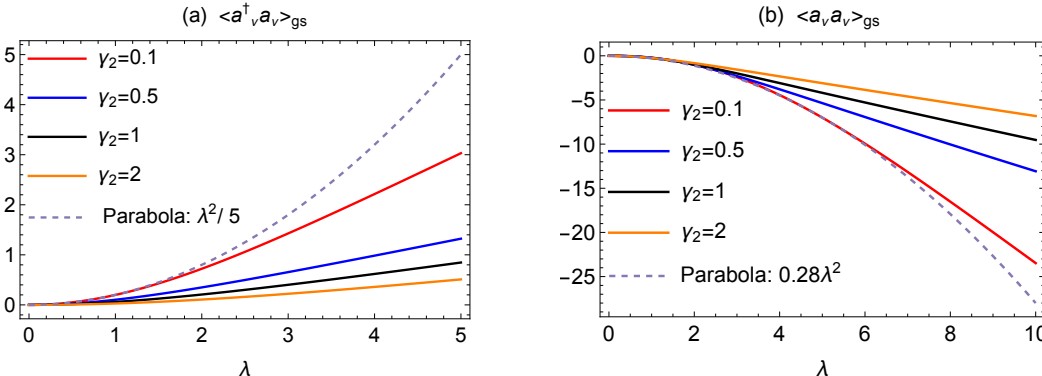

Figure 6: (a) Ground state photon occupation $\langle \hat{a}_\nu^\dagger \hat{a}_\nu \rangle_{gs}$ and (b) two-point photon function $\langle \hat{a}_\nu \hat{a}_\nu \rangle_{gs}$ with respect to the light-matter coupling $\lambda \sim \sqrt{N}$ for several values of the relative cavity frequency $\gamma_2 = \omega/\Omega$. The photon occupation increases for larger $\lambda$, while the enhanced magnitude of the two-point function indicates the amplified participation of two photon processes. In both panels a quadratic fit (dashed line) corresponding to the curves with $\gamma_2 = 0.1$ is provided.

This is a consequence of keeping in the Pauli-Fierz Hamiltonian [Eq. (1)], the diamagnetic $\mathbf{A}^2$ term. The importance of the diamagnetic term and its connection to the superradiant phase transition will be discussed in more detail in Sec. 6.

For $\lambda > 2$, the photon occupation deviates from the quadratic behavior exhibiting a linear trend [Fig. 6(a)]. This implies that in the thermodynamic limit the $\langle \hat{a}_\nu^\dagger \hat{a}_\nu \rangle_{gs} \sim \sqrt{N}$ and consequently the photon occupation per particle, $\langle \hat{a}_\nu^\dagger \hat{a}_\nu \rangle_{gs}/N$, vanishes. This behavior has also been identified in the Sommerfeld model coupled to the cavity in Ref. [73]. Another interesting feature is that for smaller values of $\gamma_2 = \omega/\Omega$ the photon occupation is larger because one matter excitation $\Omega$ results into several photonic excitations $\omega$, i.e., to create a photon costs less energy when $\omega$ is small. Finally, notice that for $\lambda \to 0$ the photon occupation is zero as expected in the decoupling limit. It is important to mention, that as long as the system is closed, the ground state photons cannot decay. However, in a lossy cavity the photons in the polaritonic ground-state will eventually leak out of the cavity and will become measurable.

Moreover, it is interesting to examine the two-point photon function $\langle \hat{a}_\nu \hat{a}_\nu \rangle_{gs} = \langle \hat{a}_\nu^\dagger \hat{a}_\nu^\dagger \rangle_{gs}$, which conveys information on two-photon excitations/de-excitations in the polaritonic ground state. We compute the two-photon processes and find

$$\langle \hat{a}_\nu \hat{a}_\nu \rangle_{gs} = \frac{1}{4 + 4\Lambda^2} \left[ \frac{\gamma_2}{\Omega_-/\Omega} - \frac{\Omega_-/\Omega}{\gamma_2} + \Lambda^2 \left( \frac{\gamma_2}{\Omega_+/\Omega} - \frac{\Omega_+/\Omega}{\gamma_2} \right) \right]. \qquad (20)$$

The two-point function is illustrated in Fig. 6(b) with respect to $\lambda$ and for different values of $\gamma_2$. It increases in magnitude for larger $\lambda$ meaning that more two-photon processes occur in the polariton ground state for stronger light-matter coupling. As in the case of the photon occupation [Fig. 6(a)] the trend of $\langle \hat{a}_\nu \hat{a}_\nu \rangle_{gs}$ is initially ($\lambda < 2$) quadratic (see also the fitted quadratic curve in Fig. 6(b) for $\gamma_2 = 0.1$) and then becomes linear. Notably, however, the rate of increase of the two-point function is larger than the one of the photon occupation and most importantly it features a quadratic trend for a much more extensive $\lambda$ interval. For example, for $\gamma_2 = 0.1$ the two-point function behaves quadratically in the region $0 \le \lambda \le 6$. These two differences on the behavior of $\langle \hat{a}_\nu^\dagger \hat{a}_\nu \rangle_{gs}$ and $\langle \hat{a}_\nu \hat{a}_\nu \rangle_{gs}$ will be proved crucial for the discussion of the photon statistics in the next subsection. It is important to highlight that the photon occupation $\langle \hat{a}_\nu^\dagger \hat{a}_\nu \rangle_{gs}$ or the two-point function $\langle \hat{a}_\nu \hat{a}_\nu \rangle_{gs}$ *on their own* do not provide information about the character, statistics or correlations of the photons. The quantity that gives insights

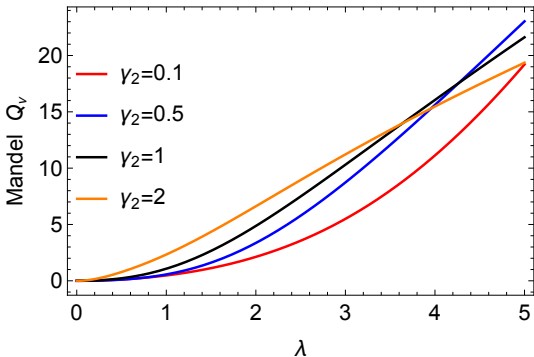

Figure 7: Mandel $Q$ parameter in terms of $\lambda$ and fixed values of $\gamma_2 = \omega/\Omega$. $Q$ is positive meaning that the photons in the polariton ground state satisfy super-Poissonian statistics, a behavior that becomes enhanced for larger $\lambda$. Crossings of $Q$ corresponding to different $\gamma_2$ at specific $\lambda$ reveal the interplay between photon occupation and the two-point function depicted in Fig. 6.

into the photon correlations and statistics is the Mandel $Q$ parameter [74] which we analyze below.

## 5.2 Photon bunching in the ground state

The Mandel $Q$ parameter measures the deviation of the photon statistics from the Poisson distribution [74]. If $-1 \leq Q < 0$, the photons follow sub-Poissonian statistics and experience an antibunching behavior, which is a feature of non-classical light. However, bunched photons are characterized by $Q > 0$ and obey super-Poissonian statistics. For $Q = 0$, photons are represented by a coherent state and obey Poisson statistics [74,76,110]. In our setting, for the polaritonic ground state $|\Psi_{gs}\rangle = \prod_\nu |0_+\rangle_\nu |0_-\rangle_\nu$, with $\langle \hat{a}^\dagger \hat{a} \rangle_{gs}$ given in Eq. (19), and the four-point function being $\langle \hat{a}_\nu^\dagger \hat{a}_\nu^\dagger \hat{a}_\nu \hat{a}_\nu \rangle_{gs} = 2\langle \hat{a}_\nu^\dagger \hat{a}_\nu \rangle_{gs}^2 + \langle \hat{a}_\nu \hat{a}_\nu \rangle_{gs}^2$ (see Appendix D for more details) the $Q$ parameter is found to be

$$Q_\nu = \frac{\langle \hat{a}_\nu^\dagger \hat{a}_\nu^\dagger \hat{a}_\nu \hat{a}_\nu \rangle_{gs} - \langle \hat{a}_\nu^\dagger \hat{a}_\nu \rangle_{gs}^2}{\langle \hat{a}_\nu^\dagger \hat{a}_\nu \rangle_{gs}} = \frac{\langle \hat{a}_\nu^\dagger \hat{a}_\nu \rangle_{gs}^2 + \langle \hat{a}_\nu \hat{a}_\nu \rangle_{gs}^2}{\langle \hat{a}_\nu^\dagger \hat{a}_\nu \rangle_{gs}} . \tag{21}$$

In the polaritonic vacuum the $Q$ parameter is strictly positive since both of its contributions for $\lambda > 0$ are finite, see Fig. 6. This implies that the ground state photons satisfy super-Poissonian statistics and thus correspond to bunched photons [74,76]. Such bunched photons due to ultrastrong light-matter coupling were also reported in Ref. [111]. Strikingly, the $Q$ parameter unlike the photon occupations and two-point function, where smaller $\gamma_2$ results in larger values for all $\lambda$, exhibits a more intricate behavior. Particularly, for different values of $\gamma_2$ crossings appear between the different "trajectories" of the $Q$ parameter in terms of $\lambda$ where before the crossing, for example $Q$ for $\gamma_2 = 2$ is larger than $Q$ for $\gamma_2 = 1$, while after the crossing the opposite holds. This phenomenon is a consequence of the competition between the photon occupation $\langle \hat{a}_\nu^\dagger \hat{a}_\nu \rangle_{gs}$ and the two-point function $\langle \hat{a}_\nu \hat{a}_\nu \rangle_{gs}$ which behave differently as a function of $\lambda$. This demonstrates that with strong and ultrastrong light-matter coupling it is possible to tailor non-trivial photon statistics and correlations in the ground state. Note that for $\lambda \to 0$ it holds $Q \to 0$, which means that in the decoupling limit photons follow trivial Poisson statistics, as expected [74,76]. It is important to mention that the positivity of the $Q$ parameter in our system holds in the polariton ground state. However, under external driving,

the excited states of the polaritons can be accessed and in this case the Mandel parameter can become negative. This would signify the generation of non-classical light [76, 112].

# 6  Superradiant Instability without the $A^2$ Term

In what follows we are interested in the importance of the often neglected [113] diamagnetic $A^2$ term whose influence on light-matter related phenomena has been studied theoretically in several publications see e.g. Refs. [73, 113–116] and its impact has also been experimentally measured in Landau polariton systems [51]. Importantly, it has been argued that eliminating the $A^2$ term leads to the well-known superradiant phase transition of the Dicke model [80]. This refers to the situation where the ground state of an ensemble of two-level systems coupled to a single quantized mode of the photon field, in the thermodynamic limit, acquires a macroscopic (infinite) photon occupation [81]. The existence though of the superradiant phase was questioned by a no-go theorem where it was shown that once the diamagnetic term is included the superradiant phase transition does not take place [117]. More recently, the possibility of a superradiant phase transition has been suggested [82, 118, 119] but again respective no-go theorems [120–122] have been derived. Lastly, the occurrence of a superradiant phase transition beyond the dipole approximation has also been suggested [123, 124].

In our case, the CM Hamiltonian of the system coupled to the cavity in the absence of the diamagnetic term is $\hat{H}'_{cm} = \hat{H}_{cm} - \frac{Ng_0^2}{2m}\hat{A}^2$. The CM Hamiltonian $\hat{H}'_{cm}$ without the $A^2$ term can be diagonalized following the procedure described in Sec. 3 with the only difference being that in the absence of the $\hat{A}^2$ term the bare cavity frequency $\omega$ *does not get renormalized* by the diamagnetic frequency $\omega_d$. Then, we obtain the respective polariton modes [see also Eq. (11) and below]

$$\Omega'_{\pm} = \sqrt{\frac{1}{2}\left(\omega^2 + \Omega^2 \pm \sqrt{4\omega_d^2\Omega^2 + (\omega^2 - \Omega^2)^2}\right)}. \tag{22}$$

Without the $\hat{A}^2$ contribution the lower polariton develops an instability for large values of the light-matter interaction. To demonstrate this, let us consider the resonance scenario $\omega = \Omega$. In this case, the lower polariton mode simplifies to $\Omega'_{-} = \sqrt{\Omega(\Omega - \omega_d)}$ which implies that for $\omega_d = \Omega$ ($\gamma_1 = 1$) the lower polariton becomes zero (or gapless) and for $\omega_d > \Omega$ it becomes imaginary signifying that the light-matter system is unstable. This instability is related to the superradiant phase transition [81]. The connection to the superradiant phase can be understood from the ground state photon occupation $\langle\hat{a}_v^\dagger\hat{a}_v\rangle_{gs}$ [see Eq. (19)] since for $\Omega'_{-} \to 0$ the photon occupation diverges, $\langle\hat{a}_v^\dagger\hat{a}_v\rangle_{gs} \to \infty$. This means that there is a macroscopic photon occupation in the ground state, i.e., a photon condensate [81, 122]. These two important consequences of neglecting the diamagnetic term are visualized in Fig. 8. Taking into account the diamagnetic term the system becomes again stable and the photon occupation is finite as it was found in Sec. 5. The fact that the system becomes unstable without the $\hat{A}^2$ term shows clearly the importance of the diamagnetic interaction which has been largely assumed that it can be neglected from the QED Hamiltonian [14, 113]. Similar conclusions were also reached for the free electron gas in a cavity [73], where it was also argued that as long as the diamagnetic $\hat{A}^2$ term is kept the system is stable and no superradiant instability occurs [73]. Our work generalizes this previous result to the case where the two-body interactions between the particles is included and the particles are bound to a scalar harmonic potential. This no-go demonstration is important because it is exact and does not rely on an asymptotic decoupling between light and matter in the thermodynamic limit and on perturbation theory [122–124].

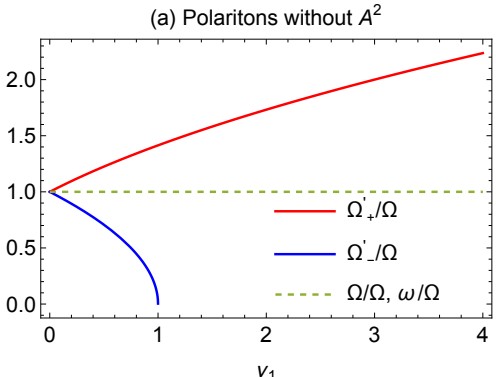
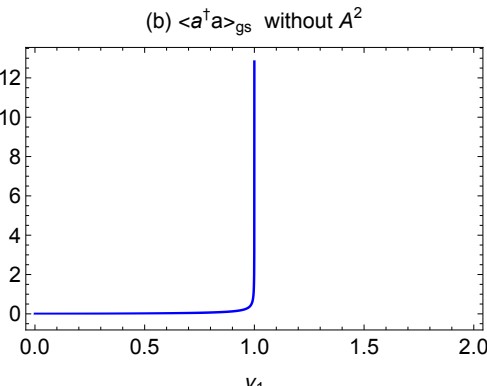

Figure 8: (a) Normalized polariton branches $\Omega'_{\pm}/\Omega$ and (b) photon occupation without the diamagnetic $\mathbf{A}^2$ term as a function of $\gamma_1 = \omega_d/\Omega$. The lower polariton develops an instability as it becomes zero at $\gamma_1 = 1$. Accordingly, the photon occupation diverges at the instability point manifesting its superradiant character. The dashed line in (a) corresponds to matter and light excitations which coincide.

# 7 Polariton-Control with a Weak Magnetic Field

Up to here we have examined the polariton properties of the system in the absence of any external perturbation. A natural question that arises concerns the influence of an external weak homogeneous magnetic field on the polariton modes. The Hamiltonian $\hat{H}_B$ including the magnetic field is $\hat{H}_B = \hat{H} + \sum_{i=1}^{N} \frac{g_0}{m}\left(i\hbar\nabla_i + g_0\hat{\mathbf{A}}\right)\cdot\mathbf{A}_{\text{ext}}(\mathbf{r}_i) + \frac{g_0^2}{2m}\mathbf{A}_{\text{ext}}^2(\mathbf{r}_i)$ where $\hat{H}$ denotes the Hamiltonian without the magnetic field [Eq. (1)] for the single-mode case. The additional terms account for the vector potential $\mathbf{A}_{\text{ext}}(\mathbf{r}) = -\mathbf{e}_x B y$ which induces a homogeneous magnetic field in the $z$ direction. The polaritons in the original Hamiltonian $\hat{H}$ form in the CM frame. For that purpose we transform the total Hamiltonian in the CM and relative coordinates frames. Then, the Hamiltonian reads

$$\hat{H}_B = \hat{H}_{\text{cm}} + \frac{g_0}{m}\left(i\hbar\nabla_{\mathbf{R}} + g_0\sqrt{N}\hat{\mathbf{A}}\right)\cdot\mathbf{A}_{\text{ext}}(\mathbf{R}) + \frac{g_0^2}{2m}\mathbf{A}_{\text{ext}}^2(\mathbf{R}) + \hat{H}_{\text{rel}}(\{\mathbf{R}_j, \mathbf{A}_{\text{ext}}(\mathbf{R}_j)\}). \quad (23)$$

For the polariton states the relative degrees of freedom, $\mathbf{R}_j$ with $j > 1$, are irrelevant because they are decoupled from the CM part. Thus, we can neglect $\hat{H}_{\text{rel}}$. To find the effect of the magnetic field on the polariton states we express the $x$-component of the momentum operator $\nabla_{\mathbf{R}}$, the quantized photon field $\hat{\mathbf{A}}$ and the magnetic field $\mathbf{A}_{\text{ext}}$ in terms of the polaritonic operators $\{\hat{d}_{x+}, \hat{d}_{y+}, \hat{d}_{x-}, \hat{d}_{y-}\}$, see Appendix C. The strength of the external magnetic field is considered to be weak as compared to the frequency of the trapping potential $\Omega$. This is quantified by the ratio $\omega_B/\Omega$, where $\omega_B = g_0 B/m$ is the magnetic-field dependent frequency. Thus, the contribution of the magnetic field can be treated perturbatively. To first order in perturbation theory the bilinear term $\left(i\hbar\nabla_{\mathbf{R}} + g_0\sqrt{N}\hat{\mathbf{A}}\right)\cdot\mathbf{A}_{\text{ext}}(\mathbf{R})$ does not contribute to the shift of the polaritonic energy levels. This coupling term involves only scatterings between the polaritons in the different directions of the form $\sim \hat{d}_x\hat{d}_y$ whose expectation value vanishes in the polariton ground state. Thus, only the diamagnetic term $\mathbf{A}_{\text{ext}}^2$ modifies the polariton energy levels, which read

$$\delta E_{n_+,n_-} = \frac{g_0^2}{2m}\,_y\langle n_+|_y\langle n_-|\mathbf{A}_{\text{ext}}^2|n_-\rangle_y|n_+\rangle_y = \frac{\hbar\omega_B^2}{2\Omega^2}\left[\frac{\Omega_+}{1+\Lambda^2}\left(n_+ + \frac{1}{2}\right) + \frac{\Lambda^2\Omega_-}{1+\Lambda^2}\left(n_- + \frac{1}{2}\right)\right], \quad (24)$$

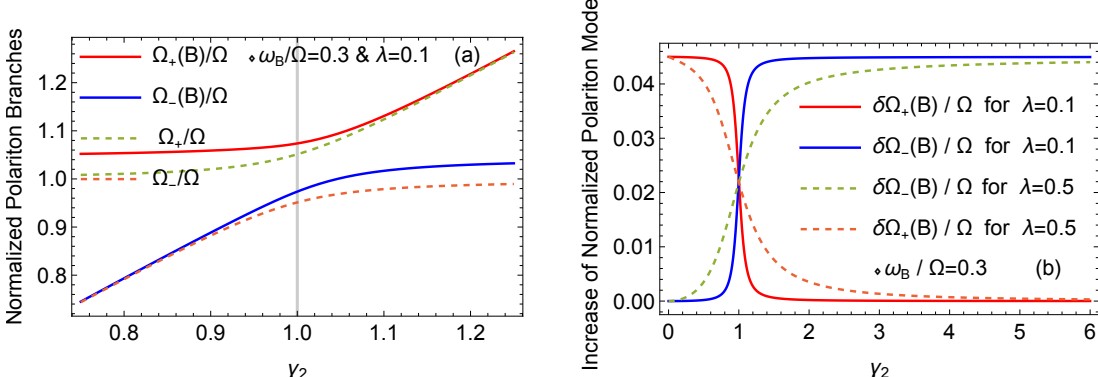

Figure 9: (a) Polariton branches and (b) energy increase of each branch in the presence of an external magnetic field $B$ with respect to $\gamma_2 = \omega/\Omega$. Dashed lines in (a) illustrate the polariton branches in the absence of $B$, while in (b) refer to different values of $\lambda$ (see legend). The magnetic field shifts the point of the avoided crossing (minimum energy difference) and the polariton gap decreases (panel (a)). In (b) the polaritons exchange energy as a function of $\gamma_2$ and at $\gamma_2 = 1$ they possess exactly the same energy increase.

with $\omega_B = g_0 B/m$. The above result gives the correction to the polariton energy levels due to the external magnetic field

$$\delta\Omega_+(B) = \frac{\Omega_+ \omega_B^2}{2\Omega^2(1+\Lambda^2)} \quad \text{and} \quad \delta\Omega_-(B) = \frac{\Omega_- \omega_B^2 \Lambda^2}{2\Omega^2(1+\Lambda^2)}. \tag{25}$$

Therefore, the polariton modes under the external perturbation become $\Omega_+(B) = \Omega_+ + \delta\Omega_+(B)$ and $\Omega_-(B) = \Omega_- + \delta\Omega_-(B)$. The polariton branches for varying $\gamma_2$ are shown in Fig. 9(a). It is evident that within $\gamma_2 < 1$, i.e. before the avoided crossing, the energy of the magnetic field is absorbed by the upper polariton while the lower one remains unaffected, see the deviation of each branch from its bare excitation energy. In the vicinity of the avoided crossing, $\gamma_2 = 1$, the polariton branches exchange energy and for $\gamma_2 > 1$ the lower polariton acquires the energy of the upper polariton which turns back to its original unperturbed value. Thus, the magnetic field facilitates energy transfer between the two polariton states.

To further analyze this energy transfer process we track separately the corrections of the normalized polariton branches $\delta\Omega_+(B)/\Omega$ and $\delta\Omega_-(B)/\Omega$ as a function of $\gamma_2$, see Fig. 9(b). For $\gamma_2 < 1$ the energy of the magnetic field is absorbed by the upper polariton, while increasing $\gamma_2$ towards the resonance point ($\gamma_2 = 1$) the upper polariton transfers its energy to the lower one. At the resonance point the two branches $\delta\Omega_\pm(B)/\Omega$ coincide, meaning that the upper polariton has transferred half of its energy to the lower one. As such, the two polaritons acquire exactly the same amount of energy from the magnetic field. Beyond $\gamma_2 = 1$ the energy of the upper polariton continues to decrease and eventually all the energy is transferred to the lower polariton. At resonance $\delta\Omega_+(B) = \delta\Omega_-(B)$ independently of the value of the light-matter coupling, as it can be seen from Fig. 9(b), while the rate of the energy exchange depends on $\lambda$. These findings pave the way for future investigations devoted to unravel the interplay of interparticle correlations and the energy transfer among the polaritons, especially so by devising specific dynamical protocols.

## 7.1 Behavior of the polariton gap

In addition to the inter-polariton energy exchange there are several other important phenomena that exclusively take place due to the magnetic field. As it can be seen in Fig. 9(a), the

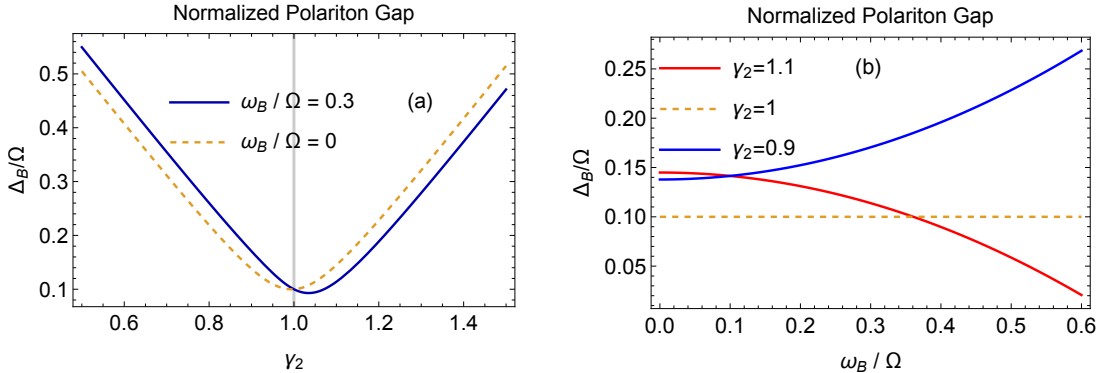

Figure 10: Normalized polariton gap $\Delta_B/\Omega$ for $\lambda = 0.1$ in the presence of a weak external magnetic field. (a) $\Delta_B/\Omega$ with respect to $\gamma_2$ and (b) in terms of $\omega_B/\Omega$. In both cases the polariton gap at resonance $\gamma_2 = 1$ is not affected by the magnetic field while beyond the resonance point, and for particular values of the magnetic field, becomes smaller than the gap at resonance.

polariton gap closes due to the magnetic field, compare with the gap among the bare excitation energies depicted with the dashed lines. The key observation is that the point at which the polaritons actually come to the closest proximity is no longer the resonance point $\gamma_2 = 1$. This can be understood through the energy gap $\Delta_B = \Omega_+(B) - \Omega_-(B)$ between the polaritons as a function of the magnetic field

$$\frac{\Delta_B}{\Omega} = \frac{\Omega_+ - \Omega_-}{\Omega} + \frac{\omega_B^2}{2\Omega^2(1+\Lambda^2)}\left(\frac{\Omega_+}{\Omega} - \Lambda^2\frac{\Omega_-}{\Omega}\right). \tag{26}$$

The first term refers to the gap $\Delta$ for zero magnetic field and only the second contribution depends on the strength of the magnetic field $B$ through $\omega_B$. Particularly, Fig. 10(a) depicts the normalized gap $\Delta_B/\Omega$ as a function of $\gamma_2$ with fixed magnetic field $(\omega_B/\Omega)$ and in Fig. 10(b) we fix $\gamma_2$ showcasing the gap in terms of $\omega_B/\Omega$. In both cases the value of the gap at the resonance point $\gamma_2 = 1$ is not affected by the magnetic field. This is true because at $\gamma_2 = 1$ the second term in Eq. (26) vanishes. Moreover, we readily observe that beyond the resonance point, i.e., for $\gamma_2 > 1$, the polariton gap in the presence of the magnetic field can become smaller as compared to $\gamma_2 = 1$. This does not occur for all values of the magnetic field but only after a particular critical value of $\omega_B/\Omega$ as shown in Fig. 10(b). This non-trivial dependence of the polariton gap to the magnetic field strength is important for the associated Landau-Zener transition probability [83] which we discuss below.

## 7.2 Landau-Zener transition

The width of the avoided crossing can be manipulated by the external magnetic field and thus it also influences the probability of a diabatic transition in the vicinity of the avoided crossing between the polariton branches. This probability is given by the well-known Landau-Zener formula [83] $P_{LZ} = e^{-2\pi\Gamma}$ with $\Gamma = \frac{\Delta^2}{\hbar|\nu|}$. Also, $\Delta$ is half of the energy difference between the two levels at the avoided crossing, $2\Delta_B = \hbar(\Omega_+(B) - \Omega_-(B))$, and $\nu$ is the Landau-Zener velocity dictating the rate at which the crossing is traversed. We utilize the Landau-Zener formula to infer the probability of a diabatic transition from the lower to the upper polariton, while varying the cavity frequency by moving slowly the cavity mirrors. To illustrate the magnetic field dependence of the transition probability, we introduce the relative polariton frequencies $\Omega_+(B)/\Omega$, $\Omega_-(B)/\Omega$ obtaining $P_{LZ} = e^{-\frac{\pi\hbar\Omega^2}{2|\nu|}\left(\frac{\Delta_B}{\Omega}\right)^2}$. The respective Landau-Zener probability as

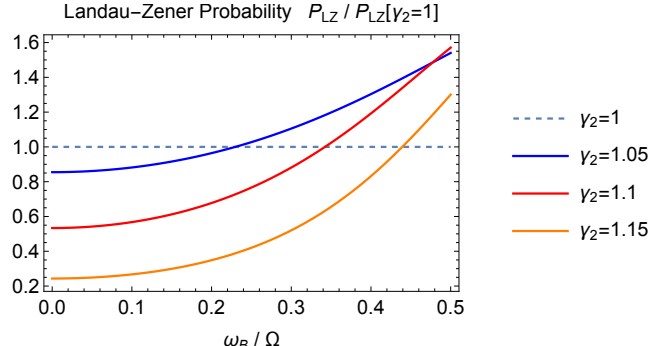

Figure 11: Landau-Zener probability as a function of $\omega_B/\Omega = g_0 B/m\Omega$ with $\lambda = 0.1$, $\hbar\Omega^2/2|v| = 2$ and for different values of $\gamma_2$. For $\gamma_2 > 1$, i.e. beyond the resonance point, there is a critical value of the magnetic field at which the transition probability becomes larger than the one at resonance $\gamma_2 = 1$.

a function of the dimensionless ratio $\omega_B/\Omega$, normalized by the Landau-Zener probability at resonance $P_{LZ[\gamma_2=1]}$, is provided in Fig.11. The Landau-Zener probability at the resonance point $\gamma_2 = 1$ is independent of $\omega_B/\Omega$, because the polariton gap $\Delta_B$ is unaffected by the magnetic field at $\gamma_2 = 1$ as shown in Fig. 10(b). Turning to $\gamma_2 > 1$ [Fig. 11], there is a critical magnetic field strength above which the Landau-Zener probability becomes larger than the one at resonance. This observation implies that an external magnetic field can be used to control the point at which a Landau-Zener transition between the polaritons takes place. In this sense, there is an interesting interplay between the polariton states due to the external magnetic field, which can be utilized to tune some of their fundamental properties, as well as generate an exchange of energy between them.

## 8 Summary and Outlook

We study the formation of collective polariton states emerging in the ground state of a harmonically trapped many-particle interacting system, strongly coupled to a spatially homogeneous cavity field. We demonstrate that the light field couples to the particle CM, while it decouples from the interparticle interactions. As such, it is possible to analytically obtain the exact collective polariton states and describe various light-induced phenomena, like the increase of the effective mass of the particles, effective interactions mediated in the matter subsystem by the cavity field and the behavior of photon correlations.

By inspecting the matter subsystem we exemplify that the cavity field enhances the spatial localization of the CM density profile becoming more prominent in the ultrastrong coupling regime. This localization phenomenon is related to the increase of the effective mass of the particles in the polaritonic ground state and it is further corroborated by deriving a corresponding effective potential picture mediated by the cavity field into the matter subsystem. The latter yields an enhanced external trap for the matter field but most importantly induces long-range pairwise interparticle interactions. Utilizing the effective matter Hamiltonian we compute the ground-state density of the matter subsystem showing that its density exhibits a similar enhancement of localization with the CM which increases with the number of particles coupled to the cavity. Thus, we observe a collective, Dicke-type, phenomenon in the ground state of the many-particle system. This is an important finding demonstrating that Dicke-collectivity can exist also in the ground state of many-particle systems coupled to the photon field. This elevates Dicke-collectivity from being an excited-state property in the spontaneous emission

(Dicke superradiance) [80], to a ground state phenomenon as well.

Moreover, the cavity-enhanced localization of the ground state exhibits a maximum on resonance. This could potentially provide insights on the resonantly-modified ground state chemical properties observed in polaritonic chemistry [5–7,32,77,78]. The polaritonic ground state, being a correlated state between photons and matter, contains a non-trivial photon population [73]. Particularly, due to the matter-mediated correlations, the photons obey super-Poissonian statistics implying that they are bunched in the ground state [76,110,111,125].

Turning to the impact of the often neglected diamagnetic interactions we showcase that if the $\mathbf{A}^2$ term is neglected the system develops an instability. The latter manifests by the fact that the lower polariton at a critical value of the light-matter interaction goes to zero, and beyond this critical point becomes imaginary signifying that the Hamiltonian is unstable. This is a behavior similar to the one discussed in Ref. [82] for the Hopfield model [126]. At the critical point the ground-state photon occupation diverges, which means that photon condensation occurs [122,124], and thus the instability is of superradiant character [81]. However, as long as the diamagnetic term is included, the light-matter system is stable and the superradiant phase transition is prevented.

Upon considering an external perturbation, through a homogeneous magnetic field, we reveal that it substantially affects the properties of the polariton branches. Indeed, it induces a coherent energy transfer among the polaritons which acquire exactly the same amount of energy at resonance. Accordingly, the polariton gap at resonance is insensitive to magnetic field variations, but outside the resonance it is affected. Namely, below (above) resonance the gap is larger (smaller) than at resonance. This phenomenon has direct implications on the respective Landau-Zener transition probability [83] between the polaritons which is enhanced via the magnetic field beyond the resonance point.

Our work provides analytical insights to strong and ultrastrong light-matter interactions and paves the way for several future directions aiming to reveal polariton phenomena in many-body cavity QED settings. An interesting possibility is to generalize our treatment, introduced in Sec. 2.1, in order to study collective polariton states, in multi-mode cavities which are currently of intense theoretical and experimental interest [109,127]. Employing a homogeneous time-dependent electric field it would be possible to probe the dynamical formation of polariton states and in general monitor their non-equilibrium time-evolution. Certainly, a deeper understanding of the cavity mediated interactions and their competition with direct two-body interactions are worth to be pursued, aiming in particular to address whether collectivity still survives. Additionally, the consideration of a two-component system inside the cavity will give rise to several interesting phenomena. For instance, it is expected that the presence of two-body inter-component coupling will facilitate the generation of mediated interactions between the polariton quasiparticles of the different components [31]. These polariton-polariton interactions [128,129] will introduce polariton non-linearities [130] which could potentially lead to polariton condensation [131]. Finally, polariton-polariton interactions can also emerge, even for a single component system, e.g. by considering an inhomogeneous cavity field.

## Acknowledgements

We would like to thank Dan Stamper-Kurn for fruitful discussions. The authors acknowledge support from the NSF through a grant for ITAMP at Harvard University. S. I. M. was also supported in part by the National Science Foundation under Grant No. NSF PHY-1748958.

## A  Relative part of the Hamiltonian

In Sec. 2.1 we showed how the CM and the cavity degrees of freedom separate from the relative coordinates, and we gave the expression for the CM part of the Hamiltonian. For completeness here we provide also how the relative part of the Hamiltonian that depends on $\{\mathbf{R}_j\}$ with $j > 1$. Particularly, it takes the form

$$\hat{H}_{\text{rel}} = \frac{1}{2m} \sum_{j=2}^{N} \left( \frac{i\hbar}{\sqrt{N}} \nabla_{\mathbf{R}_j} \right)^2 - \frac{\hbar^2}{2mN} \sum_{j,k=2}^{N} \nabla_{\mathbf{R}_j} \cdot \nabla_{\mathbf{R}_k} + \frac{m\Omega^2}{2} N \sum_{j=2}^{N} \mathbf{R}_j^2 - \frac{m\Omega^2}{2} \left( \sum_{j=2}^{N} \mathbf{R}_j \right)^2$$
$$+ \sum_{1<l}^{N} W(\sqrt{N}|\mathbf{R}_l|) + \sum_{2 \leq i < l}^{N} W(\sqrt{N}|\mathbf{R}_i - \mathbf{R}_l|) \,. \tag{A.1}$$

## B  Exact Solution in Free Space

In this Appendix we focus on the case where the matter subsystem lies in free space without an external trap, i.e., $\Omega = 0$. As we already explained within the main text in Sec. 2, the relative coordinate part of the Hamiltonian does not couple to the quantized light field. As a consequence we only focus on the CM part of the Hamiltonian which in the single-mode case considered throughout [see Eq. (6)] reads

$$\hat{H}_{\text{cm}} = -\frac{\hbar^2}{2m} \nabla_{\mathbf{R}}^2 + i g_0 \hbar \sqrt{N} \hat{\mathbf{A}} \cdot \nabla_{\mathbf{R}} + \frac{N g_0^2}{2m} \hat{\mathbf{A}}^2 + \underbrace{\sum_{\nu=x,y}^{2} \hbar\omega \left[ \hat{a}_\nu^\dagger \hat{a}_\nu + \frac{1}{2} \right]}_{\hat{H}_p} \,. \tag{B.1}$$

In the above expression $\hat{H}_p$ solely depends on the photon annihilation and creation operators i.e. $\{\hat{a}_\nu, \hat{a}_\nu^\dagger\}$. As argued in Sec. 3 this Hamiltonian can be brought into a diagonal form by defining the bosonic operators $\hat{b}_\nu = \frac{1}{2\sqrt{\omega\widetilde{\omega}}} \left[ \hat{a}_\nu(\widetilde{\omega} + \omega) + \hat{a}_\nu^\dagger(\widetilde{\omega} - \omega) \right]$ (and its conjugate $\hat{b}_\nu^\dagger$). Note, that this transformation is equivalent to the scaling transformation we performed in Sec. 3 on the photonic displacement coordinates. Accordingly, the Hamiltonian takes the form

$$\hat{H}_{\text{cm}} = -\frac{\hbar^2}{2m} \nabla_{\mathbf{R}}^2 + i g \sum_{\nu=x,y} \mathbf{e}_\nu \left( \hat{b}_\nu^\dagger + \hat{b}_\nu \right) \cdot \nabla_{\mathbf{R}} + \sum_{\nu=x,y}^{2} \hbar\widetilde{\omega} \left( \hat{b}_\nu^\dagger \hat{b}_\nu + \frac{1}{2} \right) \,. \tag{B.2}$$

Recall that $g = \omega_d \sqrt{\hbar^3/2m\widetilde{\omega}}$ is the collective light-matter coupling constant. The Hamiltonian of Eq. (B.2) is invariant under translations in the matter configuration space, since it only includes the momentum operator of the particles. This implies that $\hat{H}_{\text{cm}}$ commutes with the momentum operator $\nabla$, $[\hat{H}_{\text{cm}}, \nabla_{\mathbf{R}}] = 0$, and the eigenfunctions of the CM are plane waves of the form $\phi_{\mathbf{K}} = e^{i\mathbf{K}\cdot\mathbf{R}}/\sqrt{\mathcal{V}}$. Applying the Hamiltonian $\hat{H}_{\text{cm}}$ on the eigenfunction $\phi_{\mathbf{K}}$ we have

$$\hat{H}_{\text{cm}} \phi_{\mathbf{K}} = \left[ \sum_{\nu=x,y}^{2} \left[ \hbar\widetilde{\omega} \left( \hat{b}_\nu^\dagger \hat{b}_\nu + \frac{1}{2} \right) - g \left( \hat{b}_\nu + \hat{b}_\nu^\dagger \right) \mathbf{e}_\nu \cdot \mathbf{K} \right] + \frac{\hbar^2 \mathbf{K}^2}{2m} \right] \phi_{\mathbf{K}} \,. \tag{B.3}$$

Defining now another set of annihilation and creation operators $\{\hat{c}_\nu^\dagger, \hat{c}_\nu\}$

$$\hat{c}_\nu = \hat{b}_\nu - \frac{g \mathbf{e}_\nu \cdot \mathbf{K}}{\hbar\widetilde{\omega}} \quad \text{and} \quad \hat{c}_\nu^\dagger = \hat{b}_\nu^\dagger - \frac{g \mathbf{e}_\nu \cdot \mathbf{K}}{\hbar\widetilde{\omega}} \,, \tag{B.4}$$

the operator $\hat{H}_{\mathrm{cm}}\phi_{\mathbf{K}}$ given by Eq. (B.3) simplifies as follows

$$\hat{H}_{\mathrm{cm}}\phi_{\mathbf{K}} = \left[ \sum_{\nu=x,y}^{2} \left[ \hbar\widetilde{\omega}\left( \hat{c}_{\nu}^{\dagger}\hat{c}_{\nu} + \frac{1}{2} \right) - \frac{g^2}{\hbar\widetilde{\omega}}(\mathbf{e}_{\nu}\cdot\mathbf{K})^2 \right] + \frac{\hbar^2}{2m}\mathbf{K}^2 \right]\phi_{\mathbf{K}}.$$

The operators defined in Eq. (B.4) also satisfy bosonic commutation relations $[\hat{c}_{\nu},\hat{c}_{\nu'}^{\dagger}] = \delta_{\nu\nu'}$ for $\nu,\nu' = x,y$. For the operator $\hat{c}_{\nu}^{\dagger}\hat{c}_{\nu}$ the full set of eigenstates is [87]

$$|n_{\nu},\mathbf{e}_{\nu}\cdot\mathbf{K}\rangle = \frac{(\hat{c}_{\nu}^{\dagger})^{n_{\nu}}}{\sqrt{n_{\nu}!}}|0_{\nu},\mathbf{e}_{\nu}\cdot\mathbf{K}\rangle, \quad \text{with} \quad n_{\nu}\in\mathbb{Z}, \ \nu=x,y, \tag{B.5}$$

where $|0_{\nu},\mathbf{e}_{\nu}\cdot\mathbf{K}\rangle$ is the ground state which gets annihilated by $\hat{c}_{\nu}$ [87], and the spectrum of the bosonic operator $\hbar\widetilde{\omega}\left( \hat{c}_{\nu}^{\dagger}\hat{c}_{\nu} + 1/2 \right)$ is $\hbar\widetilde{\omega}(n_{\nu}+1/2)$. Finally, applying $\hat{H}_{\mathrm{cm}}\phi_{\mathbf{K}}$ on the eigenstates $\prod_{\nu}|n_{\nu},\mathbf{e}_{\nu}\cdot\mathbf{K}\rangle$ of the bosonic part of the Hamiltonian we obtain

$$\hat{H}_{\mathrm{cm}}\left[ \phi_{\mathbf{K}}\prod_{\nu=x,y}^{2}|n_{\nu},\mathbf{e}_{\nu}\cdot\mathbf{K}\rangle \right] = \left( \sum_{\nu=x,y}^{2} \left[ \hbar\widetilde{\omega}\left( n_{\nu} + \frac{1}{2} \right) - \frac{g^2(\mathbf{e}_{\nu}\cdot\mathbf{K})^2}{\hbar\widetilde{\omega}} \right] + \frac{\hbar^2\mathbf{K}^2}{2m} \right)\left[ \phi_{\mathbf{K}}\prod_{\nu=x,y}^{2}|n_{\nu},\mathbf{e}_{\nu}\cdot\mathbf{K}\rangle \right]. \tag{B.6}$$

From the above expression we can deduce the exact light-matter eigenfunctions of the many-body system and the respective eigenspectrum. The solution exactly reproduces the solution obtained in Ref. [73] for the free electron gas.

## C  Matter and Photon Operators in Terms of the Polaritonic Operators

In what follows we derive the expressions for the photon and matter operators in terms of the polaritonic operators.

### C.1  Photonic operators

The annihilation and creation operators $\{\hat{a}_{\nu},\hat{a}_{\nu}^{\dagger}\}$ of the photon field in terms of the displacement coordinate $q_{\nu}$ and the conjugate momentum $\partial_{q_{\nu}}$ are

$$\hat{a}_{\nu} = \frac{1}{\sqrt{2}}\left( q_{\nu} + \frac{\partial}{\partial q_{\nu}} \right) \quad \text{and} \quad \hat{a}_{\nu}^{\dagger} = \frac{1}{\sqrt{2}}\left( q_{\nu} - \frac{\partial}{\partial q_{\nu}} \right). \tag{C.1}$$

The coordinate $q_{\nu}$ and its momentum are related to $V_{\nu-}$ and $\partial_{V_{\nu-}}$ via the relations

$$q_{\nu} = -\sqrt{\frac{\omega}{\hbar}}V_{\nu-} \quad \text{and} \quad \frac{\partial}{\partial q_{\nu}} = -\sqrt{\frac{\hbar}{\omega}}\frac{\partial}{\partial V_{\nu-}}. \tag{C.2}$$

Using the expressions $S_{\nu l} = \sum_{j}O_{jl}V_{\nu j}$ and $\partial/\partial S_{\nu l} = \sum_{j}O_{jl}\partial/\partial V_{\nu j}$, we find $V_{\nu-}$ and $\partial_{V_{\nu-}}$ in terms of $\{S_{\nu l},\partial_{S_{\nu l}}\}$

$$V_{\nu-} = \frac{S_{\nu-} - \Lambda S_{\nu+}}{\sqrt{1+\Lambda^2}} \quad \text{and} \quad \partial_{V_{\nu-}} = \frac{\partial_{S_{\nu-}} - \Lambda\partial_{S_{\nu+}}}{\sqrt{1+\Lambda^2}}.$$

Then, the photonic displacement coordinate $q_{\nu}$ and its conjugate momentum with respect to $\{S_{\nu l},\partial_{S_{\nu l}}\}$ read

$$q_{\nu} = -\frac{S_{\nu-} - \Lambda S_{\nu+}}{\sqrt{1+\Lambda^2}}\sqrt{\frac{\omega}{\hbar}} \quad \text{and} \quad \partial_{q_{\nu}} = -\frac{\partial_{S_{\nu-}} - \Lambda\partial_{S_{\nu+}}}{\sqrt{1+\Lambda^2}}\sqrt{\frac{\hbar}{\omega}}. \tag{C.3}$$

The polariton annihilation and creation operators with regard to the polariton coordinates and momenta take the form

$$\hat{d}_{\nu l} = S_{\nu l}\sqrt{\frac{\Omega_l}{2\hbar}} + \sqrt{\frac{\hbar}{2\Omega_l}}\partial_{S_{\nu l}} \quad \text{and} \quad \hat{d}_{\nu l}^\dagger = S_{\nu l}\sqrt{\frac{\Omega_l}{2\hbar}} - \sqrt{\frac{\hbar}{2\Omega_l}}\partial_{S_{\nu l}}. \tag{C.4}$$

By inverting the above equation we express $S_{\nu l}$ and $\partial_{S_{\nu l}}$ with respect to $\{\hat{d}_{\nu l}, \hat{d}_{\nu l}^\dagger\}$

$$S_{\nu l} = \sqrt{\frac{\hbar}{2\Omega_l}}\left(\hat{d}_{\nu l}^\dagger + \hat{d}_{\nu l}\right) \quad \text{and} \quad \partial_{S_{\nu l}} = \sqrt{\frac{\Omega_l}{2\hbar}}\left(\hat{d}_{\nu l} - \hat{d}_{\nu l}^\dagger\right). \tag{C.5}$$

Then, combining Eqs. (C.3) and (C.5) we obtain the expressions for the photonic operators $\{\hat{a}_\nu, \hat{a}_\nu^\dagger\}$ in terms of the polaritonic ones $\{\hat{d}_{\nu l}, \hat{d}_{\nu l}^\dagger\}$, namely

$$\hat{a}_\nu = \frac{-1}{\sqrt{2+2\Lambda^2}}\left[\frac{\omega+\Omega_-}{\sqrt{2\Omega_-\omega}}\hat{d}_{\nu-} + \frac{\omega-\Omega_-}{\sqrt{2\Omega_-\omega}}\hat{d}_{\nu-}^\dagger - \Lambda\left(\frac{\omega+\Omega_+}{\sqrt{2\omega\Omega_+}}\hat{d}_{\nu+} + \frac{\omega-\Omega_+}{\sqrt{2\omega\Omega_+}}\hat{d}_{\nu+}^\dagger\right)\right]. \tag{C.6}$$

Accordingly, the operator $\hat{a}_\nu^\dagger$ is obtained by conjugation.

## C.2 Matter operators

The purely matter contribution $\hat{H}_{\mathrm{m}}$ of the CM Hamiltonian $\hat{H}_{\mathrm{cm}}$ is a sum of two uncoupled harmonic oscillators [Eq. (6)]. This can also be written in terms of the annihilation and creation operators as follows

$$\hat{H}_m = \sum_{\nu=x,y} -\frac{\hbar^2}{2m}\frac{\partial^2}{\partial R_\nu^2} + \frac{m\Omega^2}{2}R_\nu^2 = \sum_{\nu=x,y}\hbar\Omega\left(\hat{m}_\nu^\dagger\hat{m}_\nu + \frac{1}{2}\right), \tag{C.7}$$

where the operator

$$\hat{m}_\nu = R_\nu\sqrt{\frac{m\Omega}{2\hbar}} + \frac{\partial}{\partial R_\nu}\sqrt{\frac{\hbar}{2m\Omega}}, \tag{C.8}$$

and $\hat{m}_\nu^\dagger$ its conjugate. Recall that in order to diagonalize the light-matter Hamiltonian in Sec. 3 we performed a Fourier transform on the matter coordinates. After the Fourier transformation the matter annihilation operator becomes

$$\hat{m}_\nu = \mathrm{i}\frac{\partial}{\partial K_\nu}\sqrt{\frac{m\Omega}{2\hbar}} + \mathrm{i}K_\nu\sqrt{\frac{\hbar}{2m\Omega}}. \tag{C.9}$$

Moreover, employing the relation between $K_\nu = V_{\nu+}\sqrt{\hbar^2/m\Omega^2}$ and their conjugate momenta via the chain rule we have

$$\hat{m}_\nu = \mathrm{i}\frac{\partial}{\partial V_{\nu+}}\sqrt{\frac{\hbar}{2\Omega}} + \mathrm{i}V_{\nu+}\sqrt{\frac{\Omega}{2\hbar}}. \tag{C.10}$$

Additionally, using $S_{\nu l} = \sum_j O_{jl}V_{\nu j}$ and $\partial/\partial S_{\nu l} = \sum_j O_{jl}\partial/\partial V_{\nu j}$, we find the expressions for $V_{\nu+}$ and $\partial_{V_{\nu+}}$ in terms of $\{S_{\nu l}, \partial_{V_{\nu l}}\}$ i.e.

$$V_{\nu+} = \frac{\Lambda S_{\nu-} + S_{\nu+}}{\sqrt{1+\Lambda^2}} \quad \text{and} \quad \partial_{V_{\nu+}} = \frac{\Lambda\partial_{S_{\nu-}} + \partial_{S_{\nu+}}}{\sqrt{1+\Lambda^2}}. \tag{C.11}$$

Finally, with the use of Eq. (C.5) the expressions for the matter annihilation and creation operators with respect to the polaritonic ones are obtained. In particular

$$\hat{m}_\nu = \frac{\mathrm{i}}{\sqrt{2+2\Lambda^2}}\left[\Lambda\left(\frac{\Omega_-+\Omega}{\sqrt{2\Omega\Omega_-}}\hat{d}_{\nu-} + \frac{\Omega-\Omega_-}{\sqrt{2\Omega\Omega_-}}\hat{d}_{\nu-}^\dagger\right) + \frac{\Omega_++\Omega}{\sqrt{2\Omega_+\Omega}}\hat{d}_{\nu+} + \frac{\Omega-\Omega_+}{\sqrt{2\Omega_+\Omega}}\hat{d}_{\nu+}^\dagger\right], \tag{C.12}$$

where $\hat{m}_\nu^\dagger$ can be determined through conjugation. Notice that by combining Eqs. (C.12) and (C.9) we can find the expression for the matter operators $R_\nu$ and $\partial_{R_\nu}$ in terms of the polaritonic ones.

## C.3 Photon Field, Magnetic Field and Momentum

Here, we provide the expressions for the quantized cavity field, the momentum operator of the particles and the external magnetic field which we used for the computation of the corrections to the polariton energies under the influence of the external magnetic field in Sec.7.

$$
\begin{aligned}
i\hbar\nabla_X &= -\sqrt{\frac{\hbar m\Omega}{2+2\Lambda^2}}\left[\Lambda\frac{\Omega}{\sqrt{\Omega\Omega_-}}\left(\hat{d}_{x-}+\hat{d}_{x-}^\dagger\right)+\frac{\Omega}{\sqrt{\Omega\Omega_+}}\left(\hat{d}_{x+}+\hat{d}_{x+}^\dagger\right)\right], \\
\hat{\mathbf{A}} &= \sqrt{\frac{\hbar}{2\epsilon_0 V\omega}}\frac{-1}{\sqrt{1+\Lambda^2}}\sum_{\nu=x,y}\mathbf{e}_\nu\left[\frac{\omega}{\sqrt{\omega\Omega_-}}\left(\hat{d}_{\nu-}+\hat{d}_{\nu-}^\dagger\right)+\frac{\omega}{\sqrt{\omega\Omega_+}}\left(\hat{d}_{\nu+}+\hat{d}_{\nu+}^\dagger\right)\right], \\
\mathbf{A}_{\text{ext}}(\mathbf{R}) &= \frac{-\mathbf{e}_x iB\sqrt{\hbar}}{\sqrt{2m\Omega(1+\Lambda^2)}}\left[\Lambda\sqrt{\frac{\Omega_-}{\Omega}}\left(\hat{d}_{y-}+\hat{d}_{y-}^\dagger\right)+\sqrt{\frac{\Omega_+}{\Omega}}\left(\hat{d}_{y+}+\hat{d}_{y+}^\dagger\right)\right].
\end{aligned}
\tag{C.13}
$$

The above can be deduced from the matter and photon operators in terms of the polariton operators provided previously.

# D  Computation of the Four-Point Photon Function

Here, we elaborate on the calculation of the four-point function $\langle\hat{a}_\nu^\dagger\hat{a}_\nu^\dagger\hat{a}_\nu\hat{a}_\nu\rangle_{\text{gs}}$ appearing in the Mandel $Q$ parameter. The photon operators in terms of the polaritonic ones are given by Eq. (C.6). In the four-point operator $\hat{a}^\dagger\hat{a}^\dagger\hat{a}\hat{a}$ the terms that give non-zero contribution are:

$$
\begin{aligned}
&\hat{d}_{\nu-}\hat{d}_{\nu-}\hat{d}_{\nu-}^\dagger\hat{d}_{\nu-}^\dagger,\ \hat{d}_{\nu+}\hat{d}_{\nu+}\hat{d}_{\nu+}^\dagger\hat{d}_{\nu+}^\dagger,\ \hat{d}_{\nu-}\hat{d}_{\nu-}^\dagger\hat{d}_{\nu-}\hat{d}_{\nu-}^\dagger, \\
&\hat{d}_{\nu+}\hat{d}_{\nu+}^\dagger\hat{d}_{\nu+}\hat{d}_{\nu+}^\dagger,\ \hat{d}_{\nu-}\hat{d}_{\nu-}^\dagger\hat{d}_{\nu+}\hat{d}_{\nu+}^\dagger,\ \hat{d}_{\nu-}\hat{d}_{\nu+}\hat{d}_{\nu-}^\dagger\hat{d}_{\nu+}^\dagger, \\
&\hat{d}_{\nu+}\hat{d}_{\nu-}\hat{d}_{\nu-}^\dagger\hat{d}_{\nu+}^\dagger,\ \hat{d}_{\nu+}\hat{d}_{\nu-}\hat{d}_{\nu+}^\dagger\hat{d}_{\nu-}^\dagger,\ \hat{d}_{\nu+}\hat{d}_{\nu+}^\dagger\hat{d}_{\nu-}\hat{d}_{\nu-}^\dagger, \\
&\hat{d}_{\nu-}\hat{d}_{\nu+}\hat{d}_{\nu+}^\dagger\hat{d}_{\nu-}^\dagger.
\end{aligned}
\tag{D.1}
$$

With this information we can obtain all the non-zero contributions in the four-point function

$$
\begin{aligned}
\langle\hat{a}_\nu^\dagger\hat{a}_\nu^\dagger\hat{a}_\nu\hat{a}_\nu\rangle_{\text{gs}} &= \frac{1}{(4+4\Lambda^2)^2}\Bigg[\frac{(\Omega_--\omega)^4}{\omega^2\Omega_-^2}\langle\hat{d}_{\nu-}\hat{d}_{\nu-}\hat{d}_{\nu-}^\dagger\hat{d}_{\nu-}^\dagger\rangle_{\text{gs}} \\
&+\frac{\Lambda^4(\Omega_+-\omega)^4}{\omega^2\Omega_+^2}\langle\hat{d}_{\nu+}\hat{d}_{\nu+}\hat{d}_{\nu+}^\dagger\hat{d}_{\nu+}^\dagger\rangle_{\text{gs}}+\frac{\Lambda^4(\Omega_+-\omega)^2(\Omega_++\omega)^2}{\omega^2\Omega_+^2}\langle\hat{d}_{\nu+}\hat{d}_{\nu+}^\dagger\hat{d}_{\nu+}\hat{d}_{\nu+}^\dagger\rangle_{\text{gs}}+ \\
&+\frac{\Lambda^2(\Omega_-^2-\omega^2)(\Omega_+^2-\omega^2)}{\omega^2\Omega_+\Omega_-}\left(\langle\hat{d}_{\nu-}\hat{d}_{\nu-}^\dagger\hat{d}_{\nu+}\hat{d}_{\nu+}^\dagger\rangle_{\text{gs}}+\langle\hat{d}_{\nu+}\hat{d}_{\nu+}^\dagger\hat{d}_{\nu-}\hat{d}_{\nu-}^\dagger\rangle_{\text{gs}}\right) \\
&+\frac{\Lambda^2(\Omega_+-\omega)^2(\Omega_--\omega)^2}{\omega^2\Omega_+\Omega_-}\left(\langle\hat{d}_{\nu-}\hat{d}_{\nu+}\hat{d}_{\nu-}^\dagger\hat{d}_{\nu+}^\dagger\rangle_{\text{gs}}+\langle\hat{d}_{\nu+}\hat{d}_{\nu-}\hat{d}_{\nu-}^\dagger\hat{d}_{\nu+}^\dagger\rangle_{\text{gs}}+\langle\hat{d}_{\nu+}\hat{d}_{\nu-}\hat{d}_{\nu+}^\dagger\hat{d}_{\nu-}^\dagger\rangle_{\text{gs}}\right) \\
&+\frac{\Lambda^2(\Omega_+-\omega)^2(\Omega_--\omega)^2}{\omega^2\Omega_+\Omega_-}\langle\hat{d}_{\nu-}\hat{d}_{\nu+}\hat{d}_{\nu+}^\dagger\hat{d}_{\nu-}^\dagger\rangle_{\text{gs}}+\frac{(\Omega_--\omega)^2(\Omega_-+\omega)^2}{\omega^2\Omega_-^2}\langle\hat{d}_{\nu-}\hat{d}_{\nu-}^\dagger\hat{d}_{\nu-}\hat{d}_{\nu-}^\dagger\rangle_{\text{gs}}\Bigg].
\end{aligned}
\tag{D.2}
$$

Using the bosonic algebra of the polariton operators we find the following expression for the four-point function

$$\langle \hat{a}_\nu^\dagger \hat{a}_\nu^\dagger \hat{a}_\nu \hat{a}_\nu \rangle_{gs} = \frac{1}{(4+4\Lambda^2)^2} \left[ \frac{2(\Omega_- - \omega)^4}{\omega^2 \Omega_-^2} + \frac{(\Omega_-^2 - \omega^2)^2}{\omega^2 \Omega_-^2} + \frac{2\Lambda^4 (\Omega_+ - \omega)^4}{\omega^2 \Omega_+^2} \right.$$
$$\left. + \frac{\Lambda^4 (\Omega_+^2 - \omega^2)^2}{\omega^2 \Omega_+^2} + \frac{2\Lambda^2 (\Omega_-^2 - \omega^2)(\Omega_+^2 - \omega^2)}{\omega^2 \Omega_+ \Omega_-} + \frac{4\Lambda^2 (\Omega_+ - \omega)^2 (\Omega_- - \omega)^2}{\omega^2 \Omega_+ \Omega_-} \right].$$
(D.3)

In the last step we also used that $(\Omega_\pm - \omega)(\Omega_\pm + \omega) = \Omega_\pm^2 - \omega^2$.

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
