# Peer review of "Cavity Induced Collective Behavior in the Polaritonic Ground State"

_SciPost Physics, doi:SciPost Phys. 14, 167 (2023)_

## Round 1 · Referee Report · Anonymous · 2023-2-13

Strengths
1) Problem considered seems original
2) Exact analytical results
Weaknesses
1) Long paper
2) Some misprints
3) connection with existing experiments not evident
Report
In this manuscript, the authors investigate theoretically a system consisting of charged particles in a harmonic trap, coupled to a homogeneous cavity field.
In particular, they consider a cavity field that is homogeneous and single mode, which allows them to focus on the center of mass frame of the particles. The authors then diagonalize the center-of-mass Hamiltonian and find the collective polariton eigenmodes in Section III.
Then, in Section IV, they show that in the polariton ground state, the matter particles are more localised in the centre of the harmonic trap than in the absence of the cavity field. Equivalently, this can be interpreted in term of an enhanced effective mass of the particles.
In Section V, they calculate the photon occupation and the g2 for the ground state. In section VI, by comparing the photon occupation in the ground state obtained by keeping the diamagnetic term (A^2) in the model or not, the authors highlight that the superradiance transition is forbidden by the A^2 term. This result is analogous to previous results obtained in the context of the Dicke model.
Finally, in Section VII, the authors investigate the effect of a weak static magnetic field within a perturbative approach.
To my knowledge, the problem considered in this manuscript is original and the overall results seem correct. Therefore, I believe that an improved version of the manuscript could deserve publication.
Requested changes
1) In the introduction, the first sentence of the second last paragraph "Moreover, we demonstrate that ignoring the diamagnetic A2 term prevents the system from developing a superradiant instability." seems wrong and inconsistent with section VI. This should be corrected.
2) In Eq. (6) the fourth term contains 'e' which was absent in the original Hamiltonian (1) written only in term of the charge g_0.
Also, I find the denomination 'purely photonic Hamiltonian' for H_p a bit misleading, since it contains the term N e^2/m A^2, which would be absent without matter.
3) As far as I can see, the effect of interaction between charged particles (W in Eq.1) is not investigated. In particular, I would think that if these interactions are present, this should affect the ground-state energy of the complete Hamiltonian Hcm+Hrel. I wonder how interactions between matter-particles would affect the other results presented?
4) In my opinion, the newly added Section IV. 3 is not very clear.
Figure 5 seems to show the results of a quite obscure numerical calculation, but I am not sure to understand why the authors did not use their analytical results of Eq. 15? Also, I find quite confusing that this section focuses on a 1D matter system, while the rest of the paper seems to consider a 3D one.
5) Are there existing experiments probing systems similar to the one considered here? If so, I think it should be emphasized.
Author: Vasil Rokaj on 2023-03-03 [id 3435]
(in reply to Report 1 on 2023-02-13)We would like to thank the referee for taking the time to review our manuscript and for the positive characterization of our work. We are glad that the referee finds our manuscript original, the results correct and believes that an improved version could deserve publication. We are confident that we can address and clarify all the points raised and we hope that the referee can now recommend the revised version of our manuscript for publication.
To ease reading, in the link below, we additionally provide a pdf file with a detailed reply to all the points raised by the referee and the requested changes. In the reply letter we have kept the statements of the referee in italics. Also we provide a list of the most important changes we have made as well as a version of the manuscript with the changes in red.
https://drive.google.com/file/d/1o9VqIPxGLoJHhgJXKjHDMgu41r6KXj5q/view?usp=share_link
Reply to Requested Changes
1) We thank the referee for bringing to our attention this misprint appearing in the introduction. It was indeed meant that 'by including the diamagnetic term the superradiant phase transition is prevented' as we show in Sec.~VI. In the revised version of the manuscript we have corrected this statement such that it is in accordance with our findings.
2) The term 'purely photonic' meant that the Hamiltonian $\hat{H}_p$ depends only on photonic annihilation and creation operators $\hat{a},\hat{a}^{\dagger}$. However, the referee is correct that the diamagnetic $\hat{A}^2$ term is due to the interaction of the photon field with the matter subsystem, as it can be inferred from the dependence of the diamagnetic contribution to the particle number $N$. To avoid any further confusion and misunderstanding, in the revised version of the manuscript we refrain from calling $\hat{H}_p$ as the ``purely photonic part'' and we have revised the sentence in which this term was used.
3) The total energy of the system is $E_{tot}=E_{cm}+E_{rel}$. The light-matter interaction in our system is between the center of mass (CM) and the photons and consequently only the CM energy depends on the light-matter coupling $\lambda$, $E_{cm}(\lambda)$. The ground state energy of the CM is equal to zero-point energy of the two polariton modes $E_{cm} (\lambda)=\frac{\hbar \left(\Omega_+(\lambda)+\Omega_-(\lambda)\right)}{2}$ . Looking at Fig.2 (in the manuscript) we see that the upper polariton $\Omega_+$, which is energetically dominant, grows quadratically as a function of the light-matter coupling $\lambda$. This implies that the CM energy scales quadratically as well, $E_{cm}\sim \lambda^2$. The relative part $E_{rel}$ is independent of the light-matter interaction and contributes only a constant energy shift which is proportional to the strength of the two-body interaction $W$, which is repulsive in our system i.e. $W>0$. Thus, the interaction causes only a positive shift to the ground-state energy with respect to the non-interacting case. It is important to note however, that higher-order observables such as correlations or the particle density will be affected in a non-trivial manner from the presence of interactions. The interplay between interparticle correlations and cavity mediated interactions cannot be covered adequately here, and will be the topic of an upcoming work (see also Ref.~[107] in the manuscript).
4) First, let us clarify the details of the computations presented in Sec.~4.3. The numerical calculations for the ground-state density of the system were obtained using the mean field approximation on the effective matter Hamiltonian $H_{eff}$ given by Eq.(18). Specifically, in Fig.5(a) we show the difference between the density of the matter subsystem being coupled ($\lambda \neq 0$) and uncoupled ($\lambda=0$) to the cavity field for varying particle number. It can be readily deduced from the density difference that the cavity enhances the spatial localization of the matter subsystem. Recall here that the mean-field approximation neglects all particle correlations and it is based on the assumption that all particles being bosons occupy the same single-particle state. To further understand the impact of the cavity field as imprinted on the ground-state density Fig.5(b) presents the maximal density difference which naturally occurs at the trap center, $x=0$, as a function of the particle number $N$. Different scaling behaviors are identified depending on the strength of the light-matter interaction. In short, both figures showcase the emergent collective behavior and the enhanced density modifications for increasing particle number. It is important to note that the $n_{cm}(R)$ in Eq.~(15) is not the ground-state density $n(R)$ of the many-particle system, but only the density profile of the CM wavefunction. To obtain $n(R)$ one needs to treat also the relative Hamiltonian $H_{rel}$ which cannot be solved straightforwardly analytically as there are coupling terms between the relative degrees of freedom $R_j$ (see Appendix~A for $H_{rel}$). In order to treat all the degrees of freedom we performed the mean-field calculations presented in Section~4.3.
Next, we explain the point about the dimensionality. In our model the two polarizations of the cavity mode lie in the $(x,y)$ plane, see also Eq.~(5). This makes trivial the $z$ direction as it does not couple to the cavity. In addition, the two polarizations $e_x$ and $e_y$ are perpendicular. Thus, the cavity field does not mix the different directions in the system. This also holds for the external harmonic potential which is the sum of two one-dimensional harmonic traps, namely $V_{trap}(r)=\frac{m\Omega^2}{2} r^2=\frac{m\Omega^2}{2}\left(x^2+y^2\right)$. Due to these reasons, in the absence of interparticle interactions ($W=0$) treating only the $x$ direction is sufficient because the $y$ direction is exactly identical. This fact is explained for the CM Hamiltonian $H_{cm}$ below Eq.~(8) and the argument holds the same for $H_{eff}$ in Eq.~(18).
In the revised version of the manuscript we have added in Sec.~4.3 a brief comment about the dimensionality, such that it becomes evident why studying only one dimension is sufficient.
5) We would like to thank the referee for raising this important point, which gives us the opportunity to elaborate further on the experimental efforts probing systems similar to ours. Harmonically trapped, single ions coupled to optical cavities have been explored experimentally to a great extent. This has been done with optical cavities for quantum information processing (see Refs.~[21-23] in the revised manuscript) and for light-matter interactions at the single particle level (see Refs.~[24-27] in the revised manuscript).
However, to the best of our knowledge we are not aware of experiments where harmonically trapped charged particles, either electrons or cold ions, are collectively coupled to a cavity field. Our work proposes this as platform for the exploration of strong and ultrastrong light-matter interactions in the collective regime. Our proposal is based on the intuition and experimental advances coming from the field of cavity QED materials, where both condensed matter and chemical systems are studied under strong, collective coupling to the cavity field. On the condensed matter side, experimentalists have been able to couple two-dimensional electron gases ultrastrongly to cavity photons (see Refs.~[49-52]). While in chemistry, collective coupling is probed experimentally with organic molecules in cavities (see Refs.~[5-9]).
Following the referee's suggestion, in the revised version of our manuscript we highlight the experimental works (Refs.~[21-27] in the revised version) on harmonically trapped single ions coupled to optical cavities, in the introduction and in Sec.2 after Eq.~(1) when we introduce our model.
List of Changes
i) In the Introduction and Sec.2 of the manuscript we highlight the experimental efforts on harmonically trapped single ions coupled to cavities (Refs.~[21-27] in the manuscript) as suggested by the referee.
ii) In the Introduction of the manuscript on page 4 we have corrected a misprint in a sentence with respect to the impact of the diamagnetic interactions, as it was requested by the referee.
iii) In Eq.(6) we corrected a typographic mistake which was spotted by the referee. Also below Eq.(6) we no longer use the term 'purely photonic Hamiltonian' for $\hat{H}_p$ as it was pointed out that it could potentially lead to confusion.
iv) In Sec.4.3. we added a clarification about the dimensionality point raised by the referee.

---

## Round 1 · Author Response

To ease reading in the link below we provide a pdf file with a detailed reply to all the points raised by the referees and the requested changes. Further, in the reply letter we have kept the statements of the referees in italics. Also we provide a list of the most important changes we have made as well as a version of the manuscript with the changes in red.
https://drive.google.com/file/d/113EdH-C1qQg9_EXLq4DtPVD_EF_HEmIK/view?usp=share_link

---

## Round 2 · Referee Report · Anonymous · 2023-3-17

Strengths
-Original problem
- Analytical results
- Rather well written
Weaknesses
Paper quite long
Report
In my opinion, the authors have properly replied to my previous comments.
In particular, it is now clearer for me that the authors consider the matter part of their system to be two dimensional. I think this could be mentioned at the start of the paper. From what I can see, in the present version of the manuscript, the fact that R is a two-dimensional vector is only stated below Equation 8 on page 6.
Also, I believe that there is a small misprint below Eq. 2 where the polarization index is denoted \lambda while the letter \nu is used in the equations.
Otherwise, I think the paper meets the criteria for publication.
Requested changes
See report

---

## Round 2 · Author Response

To ease reading in the link below we provide a pdf file with a detailed reply to all the points raised by the third referee and the requested changes. Further, in the reply letter we have kept the statements of the referees in italics. Also we provide a list of the most important changes we have made as well as a version of the manuscript with the changes in red.
https://drive.google.com/file/d/1o9VqIPxGLoJHhgJXKjHDMgu41r6KXj5q/view?usp=share_link

---

## Editorial Decision

published